# Shc1 cooperates with Frs2 and Shp2 to recruit Grb2 in FGF-induced lens development

Qian Wang[1†‡], Hongge Li[1†], Yingyu Mao[1†], Ankur Garg[1], Eun Sil Park[1], Yihua Wu[1], Alyssa Chow[1], John Peregrin[1], Xin Zhang[1,2]*

[1]Department of Ophthalmology, Columbia University, New York, United States; [2]Department of Pathology and Cell Biology, Columbia University, New York, United States

*For correspondence:
xz2369@columbia.edu

†These authors contributed equally to this work

Present address: ‡Department of Ophthalmology and Visual Sciences, Washington University School of Medicine, St. Louis, United States

Competing interest: The authors declare that no competing interests exist.

## eLife Assessment

This **fundamental** article significantly advances our understanding of FGF signalling, and in particular highlights the complex modifications affecting this pathway. The evidence for the authors' claims is **convincing**, combining state of the art conditional gene deletion in the mouse lens with histological and molecular approaches. This work should be of great interest to molecular and developmental biologists beyond the lens community.

**Abstract** Fibroblast growth factor (FGF) signaling elicits multiple downstream pathways, most notably the Ras/MAPK cascade facilitated by the adaptor protein Grb2. However, the mechanism by which Grb2 is recruited to the FGF signaling complex remains unresolved. Here, we showed that genetic ablation of FGF signaling prevented murine lens induction by disrupting transcriptional regulation and actin cytoskeletal arrangements, which could be reproduced by deleting the juxtambrane region of the FGF receptor and rescued by Kras activation. Conversely, mutations affecting the Frs2-binding site on the FGF receptor or the deletion of Frs2 and Shp2 primarily impact later stages of lens vesicle development involving lens fiber cell differentiation. Our study further revealed that the loss of Grb2 abolished MAPK signaling, resulting in a profound arrest of lens development. However, removing Grb2's putative Shp2 dephosphorylation site (Y209) neither produced a detectable phenotype nor impaired MAPK signaling during lens development. Furthermore, the catalytically inactive Shp2 mutation (C459S) only modestly impaired FGF signaling, whereas replacing Shp2's C-terminal phosphorylation sites (Y542/Y580) previously implicated in Grb2 binding only caused placental defects, perinatal lethality, and reduced lacrimal gland branching without impacting lens development, suggesting that Shp2 only partially mediates Grb2 recruitment. In contrast, we observed that FGF signaling is required for the phosphorylation of the Grb2-binding sites on Shc1 and the deletion of Shc1 exacerbates the lens vesicle defect caused by Frs2 and Shp2 deletion. These findings establish Shc1 as a critical collaborator with Frs2 and Shp2 in targeting Grb2 during FGF signaling.

## Introduction

The lens is an exemplary model for studying signaling pathways (*Cvekl and Zhang, 2017*). In mice, the lens placode emerges as thickened epithelia within the lateral head ectoderm at E9.5 (*Figure 1A*). It undergoes invagination to form the lens pit at E10.5, and upon separation from the surface ectoderm, progresses into the lens vesicle at E11.5. Following this, lens progenitor cells within the lens

**eLife digest** Cells communicate by releasing proteins that bind to receptors on recipient cells, triggering a cascade of events that alter the cell's behavior. A family of signaling proteins called fibroblast growth factors (FGFs) is critical for various biological processes, especially during embryonic development. While scientists have a good understanding of how FGFs reach their target cells, less is known about the series of events they activate once they bind to a receptor.

Three adaptor proteins – called Frs2, Shp2 and Grb2 – are essential for propagating the FGF signal. First, the activated receptor binds to and adds phosphate groups to Frs2, which then recruits and facilitates the phosphorylation of Shp2 and Grb2. Here, Wang, Li, Mao et al. offer fresh insights into how this complex of molecules transmit the FGF signal through cells during lens development.

First, the team genetically modified the structure and activity of FGF receptors in mice to see how this impacted the formation of their lenses. They found that the membrane-embedded portion of the receptor, which includes the binding site for Frs2, is critical for regulating the consecutive steps of lens development. However, the initial stages of lens formation could still occur when only the Frs2 binding site was mutated. Loss of Grb2 produced a similar effect, suggesting that Frs2 and Grb2 are particularly important for the later stages of lens development.

Previous studies have suggested that Shp2 acts as a bridge between Frs2 and Grb2. To test this theory, Wang, Li, Mao et al. deleted the two sites in Shp2 that are responsible for binding to Grb2 and stopped phosphorylation interactions between the two adaptors. While these changes affected embryo survival, they had only a modest impact on lens development. Further experiments revealed that another adaptor protein called Shc1 can also mediate Grb2 recruitment and activation, and may be responsible for transmitting the FGF signal later in lens development.

This study provides deeper insights into the network of signaling molecules activated by FGFs, uncovering new mechanisms and adaptors involved in this pathway. The findings suggest that the FGF signaling network is highly adaptable, with different components being required at specific stages of development. Future research expanding on this work may lead to the discovery of therapies that target specific organs affected by FGF-related disorders.

---

vesicle proliferate and migrate toward the equator of the lens, where they differentiate into lens fibers responsible for the lens's focusing power (*Lovicu and McAvoy, 2005*). Genetic modification of the FGF signaling cascade alters various lens developmental processes, including lens invagination (*Carbe and Zhang, 2011*; *Pan et al., 2006*), lens vesicle formation (*Kuracha et al., 2011*), the establishment of the transition zone (*Li et al., 2019*), lens epithelium proliferation and survival, as well as lens fiber differentiation and elongation (*Lovicu and Overbeek, 1998*; *Qu et al., 2011b*; *Robinson et al., 1995*; *Zhao et al., 2008*). Therefore, the lens provides valuable insights into the nuanced interplay of FGF signaling components during distinct stages of lens formation.

The current model of FGF signaling posits that its primary orchestration centers on the Frs2/Shp2/Grb2 complex (*Beenken and Mohammadi, 2009*; *Brewer et al., 2016*; *Eswarakumar et al., 2005*). According to this model, FGFR activation induces phosphorylation of the adaptor protein Frs2, creating a platform for recruiting Shp2 and Grb2. In conjunction with its constitutively bound partner Sos (a guanine nucleotide exchange factor), Grb2 subsequently initiates Ras/MAPK signaling (*Hadari et al., 2001*; *Ong et al., 2000*). Prior research in eye development supports this notion, revealing that Crk proteins augment Ras signaling by associating with the Frs2/Shp2/Grb2 complex (*Collins et al., 2018*; *Li et al., 2014*; *Madakashira et al., 2012*), while PI3K-AKT signaling is activated through direct Ras binding with the PI3K catalytic subunit p110 (*Wang et al., 2021*). However, unresolved questions persist regarding downstream mediators of FGF signaling. Earlier studies suggested that mice with Frs2 mutants lacking the Grb2 binding site can survive healthily, whereas those lacking the Shp2 binding site exhibit severe eye development defects and significantly reduced MAPK signaling (*Gotoh et al., 2004*). These results suggest the critical importance of Shp2 in Grb2-mediated Ras signaling, but the exact mechanism remains unclear. Additionally, Soriano and colleagues generated allelic series of *Fgfr1* and *Fgfr2* mutants disrupting the Frs2 binding sites and multiple tyrosine phosphorylation residues, both individually and in combination, yet their phenotypes proved less severe than those of the respective null mutants (*Brewer et al., 2015*; *Clark and Soriano, 2024*) These

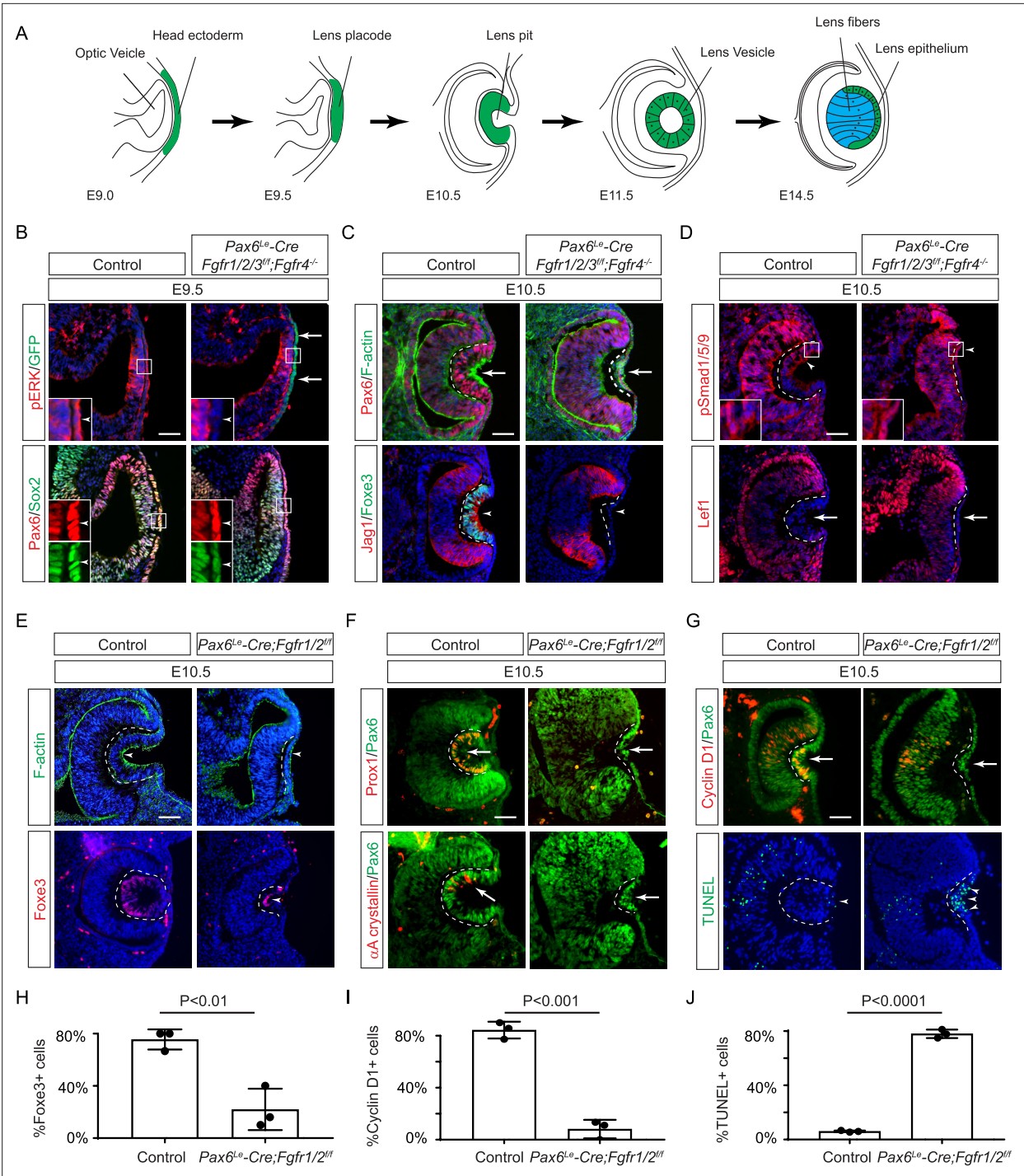

**Figure 1.** Fibroblast growth factor (FGF) signaling regulates lens development in a dose-dependent manner. (**A**) Schematic diagram of murine lens development. The head ectoderm is induced by the underlying optic vesicle to become the lens placode, which subsequently folds inwards to become the lens pit. The closure of the lens vesicle sets the stage for the differentiation of the lens epithelium into the lens fibers. (**B**) Depletion of all four *Fgfr1/2/3/4*, driven by *Pax6^Le^-Cre* and traced with GFP (arrows), led to a thinner lens placode, evident from the absence of pERK signals and the failure to upregulate Sox2 like Pax6 (inserts, arrowheads). (**C**) *Fgfr1/2/3/4* mutants displayed disrupted apical constriction (F-actin accumulation, arrows) and lacked lens-specific expression of Foxe3 and Jag1 (arrowheads). Dotted lines outline the lens pit. (**D**) Despite *Fgfr1/2/3/4* mutations, BMP (pSmad1/5/9 staining, arrowheads) and Wnt signaling (Lef1 expression, arrows) remained unaffected. (**E**) The absence of *Fgfr1/2* alone did not impede the apical buildup of F-actin nor the expression of Foxe3, indicating partial retention of lens development processes. (**F**) Crucial lens markers, Prox1 and αA-crystallin, were absent in *Fgfr1/2* mutants, pointing to a significant developmental defect after the lens induction stage. Biological replicates, n=3. (**G**) *Fgfr1/2* mutants exhibited loss of cell proliferation marker Cyclin D1 (arrows) and widespread apoptosis (TUNEL staining, arrowheads). (**H**) Quantification

*Figure 1 continued on next page*

*Figure 1 continued*

of Foxe3+ cells in *Fgfr1/2* mutants. Student's t-test, n=3, p<0.01. (**I**) Quantification of Cyclin D1+ cells in *Fgfr1/2* mutants. Student's t-test, n=3, p<0.001. (**J**) Quantification of TUNEL+ cells in *Fgfr1/2* mutants. Student's t-test, n=3, p<0.0001. Scale bars:25 μm.

The online version of this article includes the following figure supplement(s) for figure 1:

**Figure supplement 1.** Lens development in Fgf receptor mutants.

studies suggest that there may exist additional adaptor(s) other than Frs2 to mediate FGF signaling. One potential candidate for mediating FGF signaling is the Shc family of proteins, which exist in three isoforms: p66, p52, and p46 (*Wills and Jones, 2012*). These proteins contain a phosphotyrosine-binding (PTB) domain and an SH2 domain, both of which can interact with phosphorylated tyrosine residues. Additionally, Shc proteins possess multiple tyrosine residues that, upon phosphorylation, can be recognized by other molecules, including Grb2. Previous studies have demonstrated that FGF stimulation leads to increased Shc phosphorylation and its binding to Grb2, although the specific binding site remains unresolved (*Dunican et al., 2001*; *Klint et al., 1995*; *Schüller et al., 2008*). Furthermore, the functional significance of Shc's interaction with the FGF receptor in vivo has yet to be demonstrated.

In this study, we demonstrated that genetic alterations to FGFR—whether by deleting all isoforms, the juxtamembrane region, or the specific Frs2 binding site—disrupt the successive phases of lens development, encompassing lens induction, vesicle formation, and fiber differentiation. Surprisingly, while we established that Grb2-Ras signaling serves as the primary conduit of FGF signaling, interfering with Grb2 dephosphorylation and binding by Shp2 or even abolishing Shp2 phosphatase activity did not eliminate either MAPK signaling or lens differentiation. Conversely, we observed that FGF-induced Shc phosphorylation hinges on the FGFR juxtamembrane domain rather than its Frs2 binding site. Although the deletion of Shc1 has only a modest impact on lens development and MAPK activity individually, its combination with Frs2 and Shp2 deletion results in a profound arrest of lens vesicle development. These results suggest that Shc functions independently of Frs2 and Shp2 to augment Grb2-Ras signaling within the FGF pathway.

## Results
### FGF signaling is required for lens induction

Previous studies have established the presence of all four FGF receptors in the surface ectoderm during lens induction (*Garcia et al., 2011*). However, despite the deletion of the primary FGFRs, *Fgfr1*, and *Fgfr2*, only thinning of the lens placode occurred, with no discernible impact on lens determination transcription factors like Pax6, Sox2, and Foxe3. To explore whether the compensatory effects of remaining FGFRs obscure the role of FGF signaling in lens induction, we eliminated all four FGFRs using the *Pax6^Le^-Cre* deletor driven by the *Pax6* lens ectoderm (LE) enhancer (*Ashery-Padan et al., 2000*). The *Pax6^Le^-Cre* activity within the lens placode at E9.5, as indicated by the embedded GFP reporter (*Figure 1B*, arrows), resulted in the complete loss of pERK in the ectoderm, confirming FGF signaling inactivation (*Figure 1B*, inserts and arrowheads). Consequently, while the initial marker of lens induction, increased Pax6 expression as the surface ectoderm transitions into the lens placode, persisted in the mutant, Sox2 expression failed to be upregulated. Furthermore, subsequent lens placode invagination, driven by apical constriction evidenced by polarized F-actin localization on the apical side by E10.5 (*Chauhan et al., 2011*), was disrupted (*Figure 1C* arrows). This occurred despite the proper localization of Fibronectin at the basal side of the lens placode (*Figure 1—figure supplement 1A*), suggesting that overall cell polarity remained unaffected. The *Pax6^Le^-Cre;Fgfr1^f/f^;Fgfr2^f/f^;Fgfr3^f/f^;Fgfr4^-/-^* mutant also failed to exhibit the expected upregulation of Foxe3 and Jag1 in lens vesicle, further underscoring impaired lens induction (*Figure 1C*, arrowheads). Previous studies have highlighted FGF and BMP interaction during lens formation (*Garcia et al., 2011*), yet the dorsal-to-ventral gradient of BMP signaling, as indicated by pSmad staining, persisted in *Pax6^Le^-Cre;Fgfr1^f/f^;Fgfr2^f/f^;Fgfr3^f/f^;Fgfr4^-/-^* knockouts (*Figure 1D*, arrowheads). Furthermore, Wnt signaling, a known negative regulator of lens induction (*Smith et al., 2005*), showed no signs of abnormal activity, as indicated by the absence of Lef1 expression (*Figure 1D*, arrows). These results demonstrated that FGF signaling is required independently of Bmp and Wnt signaling for lens induction.

We next focused on the two primary FGFRs, Fgfr1 and 2, to scrutinize the latter phases of lens development. In contrast to mutants with deletion of all four FGFRs, we observed apical confinement of F-actin and expression of Foxe3 in the *Pax6^{Le}-Cre;Fgfr1^{f/f};Fgfr2^{f/f}* mutant lens cells, although the number of Foxe3-expressing cells were reduced (*Figure 1E and H*, arrows). However, there was a conspicuous reduction in phosphorylation of mTOR and its downstream target S6 in the lens vesicle, suggesting that deletion of *Fgfr1/2* disrupted mTOR signaling (*Figure 1—figure supplement 1B–D*, arrowheads). Underscoring the lens differentiation defect, Prox1, a crucial transcription factor for lens fiber development, along with the lens-specific protein αA crystallin expression, was also lost in the *Pax6^{Le}-Cre;Fgfr1^{f/f};Fgfr2^{f/f}* mutant lens vesicle (*Figure 1F*, arrows) (*Garg et al., 2020*; *Ochi et al., 2003*; *Xie et al., 2016*). Additionally, there was a notable decrease in Cyclin D1 expression and increasing TUNEL staining within the lens vesicle, indicating cell proliferation and apoptosis defects (*Figure 1G, I and J*, arrows and arrowheads) (*Garcia et al., 2011*).

## Ras mediates FGF signaling to suppress apoptosis in the developing lens

FGF signaling is known to stimulate the Ras-MAPK signaling pathway. To explore the significance of Ras signaling, we employed a genetic rescue strategy utilizing an inducible *Kras* allele capable of expressing the constitutively active *Kras^{G12D}* upon Cre-mediated recombination (*Figure 2A*; *Tuveson et al., 2004*). When crossed with *Fgfr1/2* mutants, this allele effectively restored pERK expression and normalized lens vesicle invagination (*Figure 2B*). By E13.5, the *Pax6^{Le}-Cre;Fgfr1^{f/f};Fgfr2^{f/f};Kras^{G12D}* lens displayed robust phosphorylation of MEK, the upstream kinase of Erk, concomitant with the expression of lens fiber markers Prox1 and αA crystallin (*Figure 2B and C*). Notably, the rescued lens reached about half the size of a control lens (*Figure 2D*). These results suggest that Kras signaling acts as a primary conduit for FGF signaling during lens development.

To ascertain whether cell death underpins the pronounced lens vesicle defect in *Pax6^{Le}-Cre;Fgfr1^{f/f};Fgfr2^{f/f}* mutant, we targeted Bax and Bak, two core regulators of the intrinsic pathway of apoptosis, for deletion. This intervention notably reduced TUNEL signals within the lens placode (*Figure 2E and F*) and resulted in a recovery of lens formation, as evidenced by the expression of Foxe3, Maf, and Jag1 (*Figure 2G*), although the lens remained considerably smaller than the wild-type control (*Figure 2H*). This underscores the critical role of FGF signaling in preventing excessive cell apoptosis during lens development.

## The juxtamembrane domain and Frs2 binding site of FGFR regulate the consecutive steps of lens development

Previous studies have mapped the Frs2 binding site to the juxtamembrane domain of FGFR (*Dhalluin et al., 2000*; *Ong et al., 2000*). To probe the role of this region in lens development, we employed two distinct alleles: one entirely devoid of the juxtamembrane domain (amino acid 407–433 in Fgfr1, *Fgfr1^{ΔFrs}*) (*Hoch and Soriano, 2006*), and another harboring mutations in two pivotal residues essential for Frs2 binding (L424A and R426A in Fgfr2, *Fgfr2^{LR}*) (*Figure 3A*; *Eswarakumar et al., 2006*). Intriguingly, while the *Fgfr1/2* compound mutant carrying *Fgfr1^{ΔFrs}* mirrored the null phenotype (*Figure 3B*, arrowheads), the mutant featuring *Fgfr2^{LR}* exhibited sustained expression of pERK, Cyclin D1, and αA crystallin, with no discernible increase in cell death, as confirmed by cleaved caspase 3 staining (see *Figure 3B and C*). This suggests that the juxtamembrane domain of FGFR likely serves additional functions beyond Frs2 binding in lens induction.

The absence of a lens induction phenotype in the *Pax6^{Le}-Cre;Fgfr1^{f/f};Fgfr2^{f/LR}* mutant raised the question regarding the role of Frs2 in lens development. Upon examining the iSyTE lens gene expression database (*Kakrana et al., 2018*), we observed a rapid increase in *Fgfr3* expression in the murine lens from E11.5 onwards, surpassing the levels of both *Fgfr1* and *Fgfr2* by E12.5 (*Figure 3D*). This led us to consider whether heightened *Fgfr3* expression could potentially mask the effects of *Fgfr2^{LR}* mutation during later stages of lens development. To explore this hypothesis, we further deleted *Fgfr3* in conjunction with the *Fgfr1* and *Fgfr2^{LR}* mutant, which indeed impeded the differentiation and elongation of posterior lens epithelial cells, as evidenced by the absence of αA crystallin, Jag1, and Maf at E11.5 (*Figure 3E*). By E12.5, unlike the *Pax6^{Le}-Cre;Fgfr1^{f/f};Fgfr2^{f/LR}* mutant, which retained pERK staining and generated lens fibers to populate the lens vesicle similar to controls, the *Pax6^{Le}-Cre;Fgfr1^{f/f};Fgfr2^{f/LR};Fgfr3^{f/f}* triple mutant remained a hollow lens vesicle without any pERK expression (*Figure 3F*

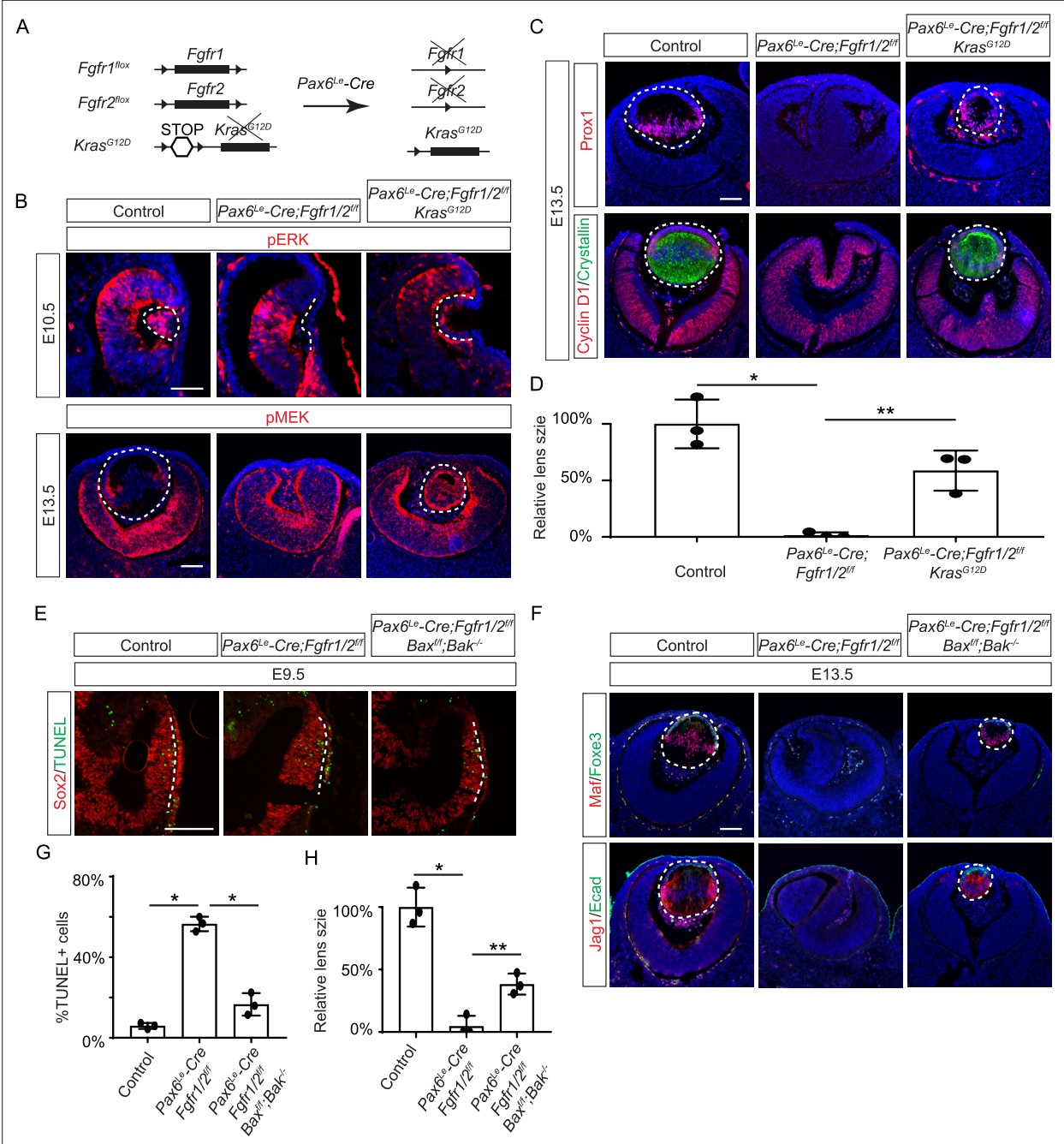

**Figure 2.** Restoration of lens development in fibroblast growth factor (FGF) signaling mutants via Kras activation and apoptosis inhibition. (**A**) The *Pax6^Le^-Cre* driver facilitated the excision of the floxed alleles of *Fgf1/2* along with the *LSL-STOP* cassette at the *Kras* locus, leading to the expression of the constitutively active *Kras^G12D^* allele within the *Fgf1/2* mutant background. (**B**) The activation of Kras signaling in the FGF signaling mutant lenses reinstated pERK activity at E10.5 and pMEK expression at E13.5, indicating restoration of MAPK signaling. (**C**) The lens-specific expression of Prox1 and αA-crystallin were also recovered, indicating successful lens development rescue. (**D**) Quantification of the lens size. One way ANOVA, n=3, *p<0.001, **p<0.02. (**E**) The deletion of pro-apoptotic genes *Bak* and *Bax* in *Fgf1/2* mutants suppressed apoptosis as shown by TUNEL staining. (**F**) Inhibiting apoptosis in *Fgfr1/2* mutants facilitated lens formation, as indicated by the expression of lens differentiation markers Prox1, Maf, and Jag1. (**G**) Quantification cell apoptosis at E9.5. One-way ANOVA, n=3, *p<0.001. (**H**) Quantification of the lens size at E13.5. One-way ANOVA, n=3, *p<0.0001, **p<0.05. Scale bars:50 μm.

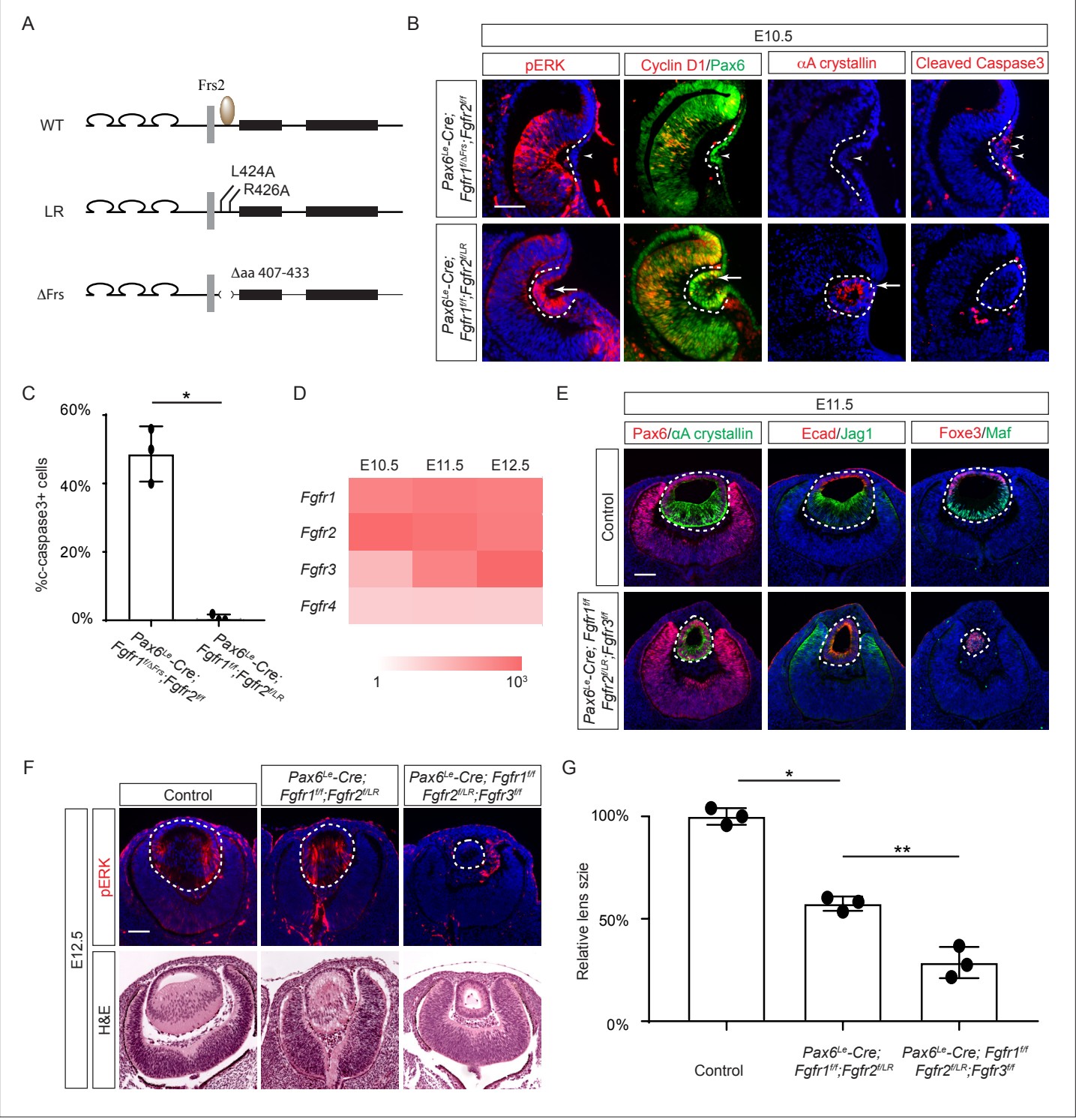

**Figure 3.** The Frs2 binding site on fibroblast growth factor receptor 2 (FGFR) is only required for lens vesicle differentiation. (**A**) Overview of *Fgfr* mutant alleles. *Fgfr1^ΔFrs* lacks the Frs2 binding domain (amino acid 407–433), and *Fgfr2^LR* has point mutations disrupting Frs2 binding (L424A and L426A). (**B**) In *Fgf1/2* compound mutants, loss of pERK, CyclinD1, αA-crystallin, and increased cleaved caspase3 were observed with the *Fgfr1^ΔFrs* allele but not the *Fgfr2^LR* allele. (**C**) Quantification of cleaved caspase3 staining. Student's t-test, n=3, *p<0.001. (**D**) Heatmap depicts *Fgfr* expression levels during lens development. (**E**) *Fgfr2^LR* mutants in the *Fgfr1/2/3* genetic background showed impaired lens vesicle differentiation, with posterior lens epithelial cells failing to elongate and activate lens fiber cell markers Jag1 and Maf. (**F**) *Fgfr1/2/3* triple mutants with *Fgfr2^LR* lost pERK staining and displayed a shallow lens vesicle at E12.5. (**G**) Quantification of the lens size. One-way ANOVA n=3, *p<0.001, **p<0.05. Scale bars:50 μm.

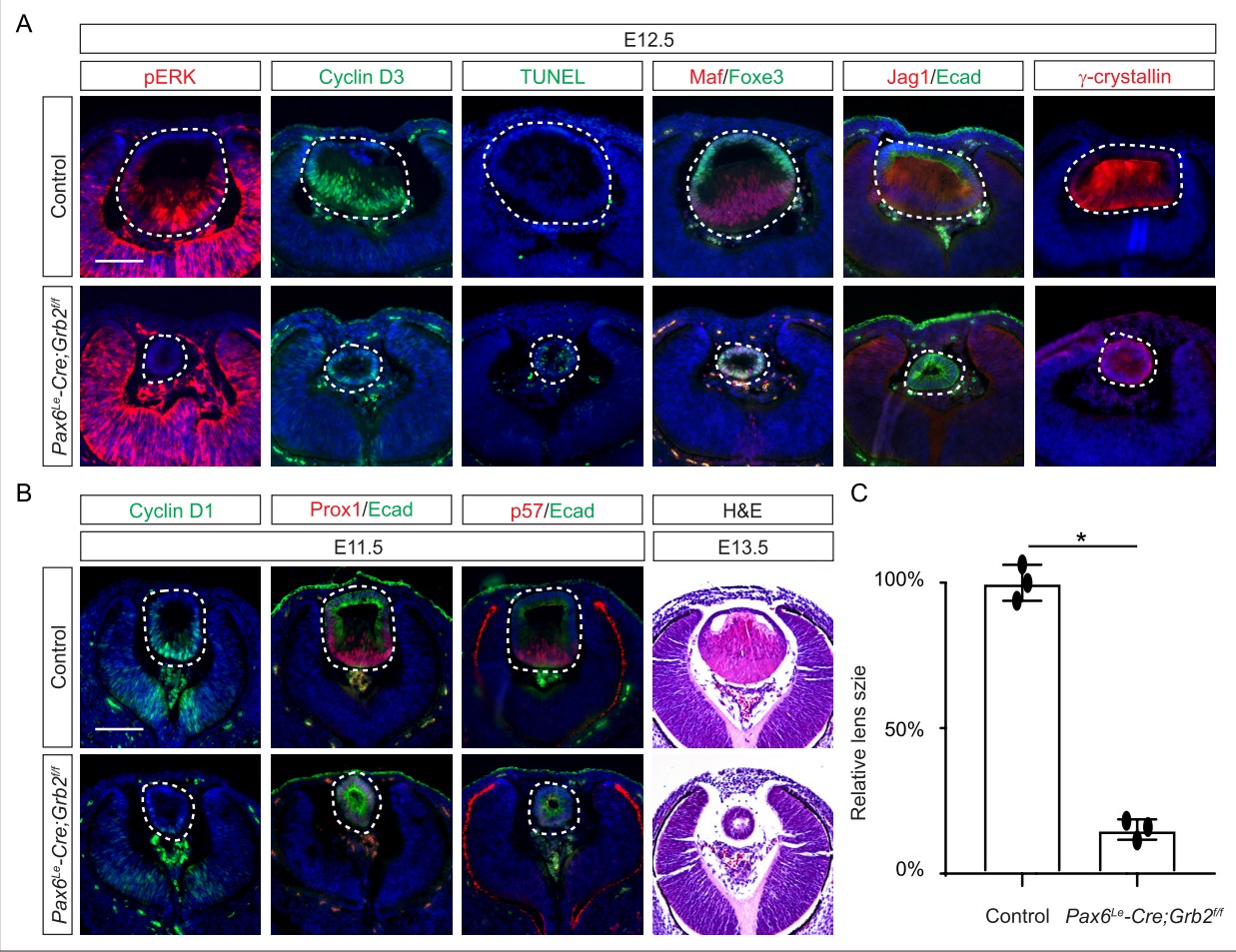

**Figure 4.** Grb2 is essential for lens vesicle survival, proliferation, and differentiation. (**A**) The targeted removal of Grb2 in the lens led to a loss of pERK signaling, reduced CyclinD3 expression, increased apoptosis (TUNEL staining), and disrupted expression of critical lens development genes Maf, Foxe3, Jag1, and γ-crystallin expression at E12.5. (**B**) Grb2 mutants displayed absent CyclinD1, Prox1, and p57 expression at E11.5 and remained an undifferentiated hollow vesicle at E13.5, failing to undergo normal lens fiber elongation. (**C**) Quantification of the lens size. Student's t-test, n=3, *p<0.0001. Scale bars:50 μm.

*and G*). This observation bears a striking resemblance to the phenotype previously reported when all three FGFRs were deleted at this stage (*Zhao et al., 2008*). Thus, while the Frs2 binding site on FGFR is dispensable for lens induction and lens vesicle formation, it evidently emerges as crucial for the later stage of lens fiber differentiation.

## Grb2 mediates FGF-MAPK signaling in lens cell differentiation

If Frs2 is responsible for lens fiber differentiation rather than lens induction, we anticipate that its downstream target, Grb2, would exhibit a similar phenotype. To test this hypothesis, we genetically deleted *Grb2* using the *Pax6^Le^-Cre* driver. Indeed, the *Pax6^Le^-Cre;Grb2^f/f^* mutant formed a lens vesicle at E12.5 but lacked pERK staining, correlating with decreased Cyclin D3 expression and increased TUNEL staining (*Figure 4A*). Notably, the initial lens determination gene Foxe3 was unaffected, but differentiation marker Jag1 was absent while Maf and crystallin were reduced. This lens differentiation defect is evident as early as E11.5, characterized by a significant reduction in the expression of cell cycle regulators Cyclin D1 and p57, as well as the pro-differentiation transcription factor Prox1, suggesting dysregulation of the cell cycle and failure to initiate the lens fiber cell differentiation program (*Figure 4B*). By E13.5, the lens vesicle remained hollow and smaller compared to the control (*Figure 4B and C*). The deletion of Grb2 affects lens differentiation without hindering the formation of the lens vesicle, mirroring the effect of Frs2 binding site mutation, thus supporting the notion that Grb2 is the primary downstream effector of Frs2 in promoting lens fiber differentiation.

## The Grb-Shp2 binding plays a modest role in FGF signaling

While both Frs2 and Shp2 possess phosphotyrosine residues that can engage with Grb2 (*Figure 5A*), it's noteworthy that only mutations in the Shp2 binding sites, rather than those in the Grb2 binding sites on Frs2, led to severe eye development defects (*Gotoh et al., 2004*). This intriguing observation spurred us to investigate the potential role of Shp2 as an adaptor in facilitating the interaction between Frs2 and Grb2. To this end, we engineered a mouse model with mutations in the Grb2 binding sites of Shp2 (*Sun et al., 2013*) by substituting the C-terminal tyrosine residues 542 and 580 with phenylalanine (*Shp2$^{YF}$*) (*Figure 5B*), which was confirmed by southern blot analysis using both 5' and 3' probes and Sanger sequencing (*Figure 5C*). However, homozygous mice carrying the *Shp2$^{YF}$* mutation died around E12.5 with visibly paler and smaller bodies (*Figure 5D and E*). Histological analysis of mutants revealed a significantly thinner labyrinth zone in placentas, crucial for oxygen and nutrient supplies (*Figure 5E*, dotted lines). To test whether this placental defect is responsible for the embryonic lethality, we combined the *Shp2$^{YF}$* mutation with the *Shp2* conditional allele using *Sox2Cre*, which is specifically active in the epiblast-derived embryonic tissues but not in the trophoblast-derived placenta (*Figure 5F*; *Hayashi and McMahon, 2002*). *Sox2Cre;Shp2$^{f/YF}$* embryos indeed survived past embryonic day 15.5 without evident morphological abnormalities, yet they succumbed shortly after birth for reasons yet unknown (*Figure 5G*).

The survival of *Sox2Cre;Shp2$^{f/YF}$* mutant beyond embryonic development permitted the isolation of mouse embryonic fibroblast (MEF) cells for biochemical analysis. As anticipated, the phosphorylation of Shp2 (pShp2$^{Y542}$) induced by FGF was abolished in *Sox2Cre;Shp2$^{f/YF}$* MEF cells, yet the activation of pERK was only partially impaired (*Figure 5H*). This contrasts with the more pronounced reduction in PDGF-induced ERK phosphorylation, underscoring a distinct requirement for Shp2-Grb2 binding across related receptor tyrosine kinase (RTK) pathways (*Araki et al., 2003*). In line with the subtle impact on FGF signaling, neither the pattern of pERK staining nor the size of the lens showed discernible alterations in *Sox2Cre;Shp2$^{f/YF}$* mutants (*Figure 5I*, arrowheads). We further investigated lacrimal gland development due to its remarkable sensitivity to FGF signaling intensity, where even a heterozygous *Fgf10* mutation has been shown to stunt gland growth (*Garg and Zhang, 2017*; *Qu et al., 2011a*). Notably, we observed a slight reduction in pERK staining in the lacrimal gland primordia at E14.5 and fewer lacrimal gland buds at birth (*Figure 5I*, arrows). These nuanced ocular phenotypes, alongside the overall normal morphology of *Sox2Cre;Shp2$^{f/YF}$* mutant embryos, collectively suggest that Shp2-Grb2 binding exerts a modest influence on FGF signaling.

## Inactivation of Shp2 phosphatase activity failed to abrogate FGF-induced MAPK signaling

The results presented above indicate that the direct binding of Grb2 to the Shp2 C-terminus is not essential for FGF signaling. This led us to explore an alternative hypothesis that Shp2 might function by removing inhibitory tyrosine phosphorylation on Grb2, thereby promoting its interaction with Sos and subsequent Ras-MAPK activation (*Ahmed et al., 2013*; *Vemulapalli et al., 2021*). Based on the PhosphoSitePlus database, we identified Y209 as the most frequently phosphorylated tyrosine residue in Grb2 (*Figure 6A*). Notably, previous studies have demonstrated that Shp2 dephosphorylates this specific site upon stimulation by various receptor tyrosine kinases (RTKs) (*Ahmed et al., 2013*; *Haines et al., 2009*; *Li et al., 2001*; *Riera et al., 2010*). To assess the functional significance of Y209 phosphorylation, we generated a mutant *Grb2* allele where Y209 was replaced with phenylalanine (*Grb2$^{YF}$*) using the ES cell-based gene targeting technique (*Figure 6B and C*). Surprisingly, our findings revealed that *Grb2$^{YF/YF}$* homozygous mutants exhibited normal viability and fertility without any obvious phenotype. Additionally, the intensity of pERK staining in mutant lenses remained unchanged in mutant lenses compared to controls, and markers of the cell cycle (Ki67 and p57) as well as differentiation (Jag1, Foxe3, and Maf) were unaffected (*Figure 6D*). These results suggest that despite being a frequent target for phosphorylation, Y209 on Grb2 is dispensable for FGF signaling.

The PhosphoSitePlus database indicates that Grb2 still possesses a less frequently phosphorylated Y160 site, which has also been previously implicated in FGF signaling (*Ahmed et al., 2015*). To rigorously assess the potential impact of Shp2-mediated Grb2 dephosphorylation, we developed a *Shp2$^{CS}$* mouse model by substituting the cysteine residue at position 463 (C459 in humans) with serine in Shp2's catalytic domain, effectively abolishing its enzymatic activity (*Figure 6E and F*). Unlike the earlier *Shp2* null mutants that perished by E7.5 (*Yang et al., 2006*), *Shp2$^{CS/CS}$* embryos exhibited

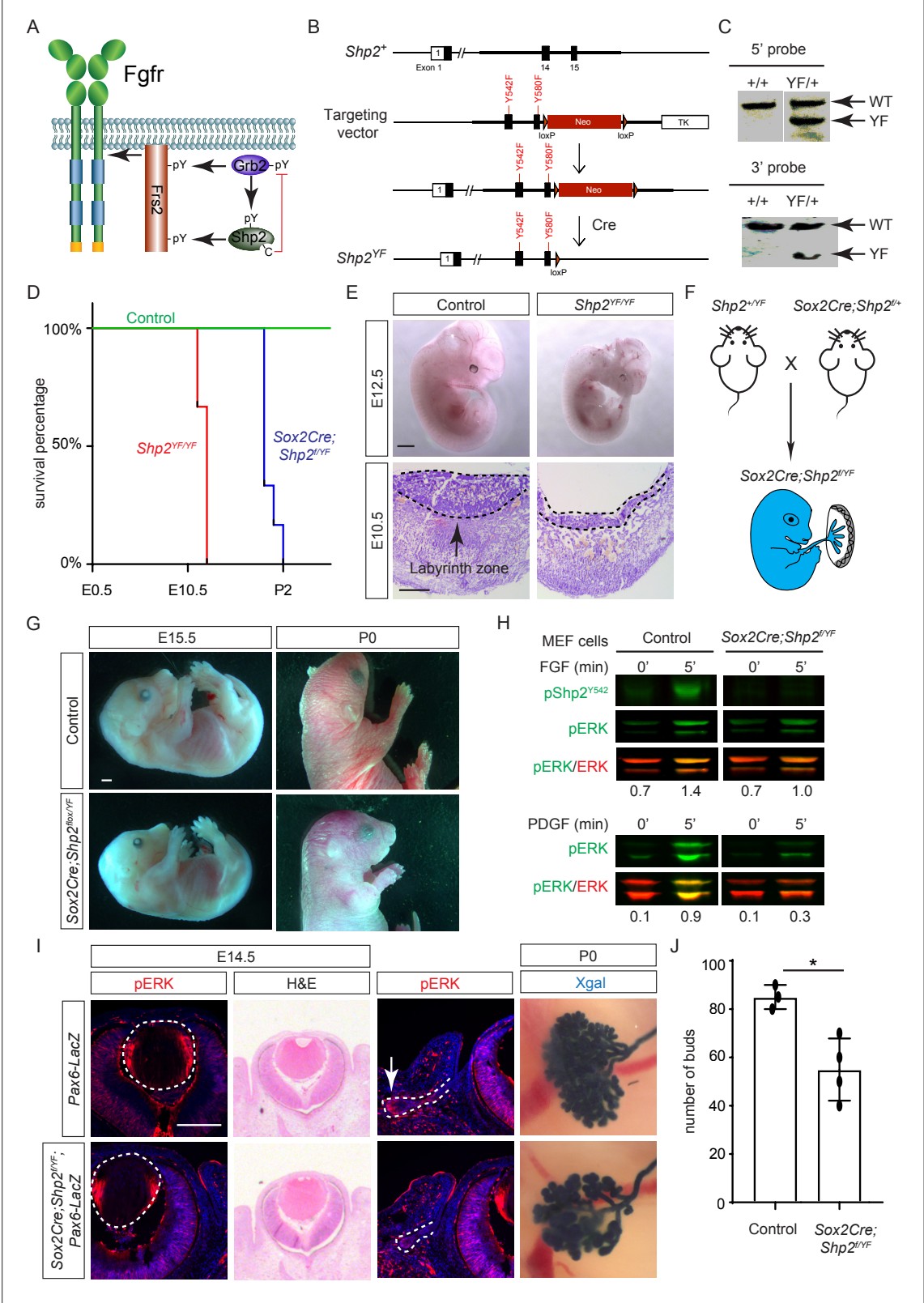

**Figure 5.** Shp2 C-terminal tyrosine phosphorylation is required for embryonic survival but dispensable for lens development. (**A**) Schematic of the core FGF signaling pathway. FGFR activation leads to phosphorylation of the adaptor Frs2 on N- and C-terminal tyrosines, recruiting Grb2 and Shp2, respectively. Shp2 can also bind Grb2 via its own C-terminal phosphotyrosines, and dephosphorylates Grb2 via its catalytic cysteine residue. (**B**) Generation of the *Shp2*^YF allele by homologous recombination to introduce *loxP*-flanked Neo and point mutations (Y542F and Y580F) disrupting the

*Figure 5 continued on next page*

*Figure 5 continued*

Shp2 C-terminal phosphotyrosine sites. The Neo cassette was subsequently excised by Cre-mediated recombination. (**C**) Validation of the *Shp2^YF^* allele targeting was confirmed through Southern blot analysis with both 5' and 3' probes. (**D**) Kaplan-Meier survival curves demonstrate early lethality of *Shp2^YF^* embryos (E12.5) and perinatal lethality of *Sox2Cre;Shp2^f/YF^* mutants. n=5 for *Shp2^YF/YF^* and n=6 for *Sox2Cre;Shp2^f/YF^* mutants. (**E**) *Shp2^YF/YF^* embryos displayed reduced body size at E12.5 and thinner labyrinth zones in their placenta at E10.5. (**F**) *Sox2Cre*-mediated targeting restricts Shp2 deficiency to the embryonic proper, circumventing placental abnormalities. (**G**) *Sox2Cre;Shp2^f/YF^* mutants appeared grossly normal at E15.5 but failed to survive after birth. (**H**) While Y542 phosphorylation in Shp2 was lost as expected, *Sox2Cre;Shp2^f/YF^* MEFs exhibited a more pronounced reduction in pERK response to PDGF stimulation compared to FGF stimulation. (**I**) *Sox2Cre;Shp2^f/YF^* mutant lens displayed normal pERK staining and morphology, but reduced pERK in lacrimal glands at E14.5 and decreased bud numbers at P0. (**J**) Quantification of the number of lacrimal gland buds. Student's t-test, n=3, *p<0.02. Scale bars: 100 μm.

The online version of this article includes the following source data for figure 5:

**Source data 1.** Original files for southern blot analysis displayed in *Figure 5C*.

**Source data 2.** Original membranes corresponding to *Figure 5*, panel C, with the relevant lanes are outlined in yellow.

**Source data 3.** Original files for western blot analysis displayed in *Figure 5H*.

**Source data 4.** Original membranes corresponding to *Figure 5*, panel H, with the relevant lanes are outlined in yellow.

stunted growth but survived until E9.5, indicating that the *Shp2^CS^* mutation doesn't entirely abrogate Shp2's function (*Figure 6G*). Moreover, after removing the *flox* allele using Cre-expressing adenovirus, the *Shp2^f/CS^* MEF cells still retained considerable pERK activity in response to FGF stimulation (*Figure 6H*). This was mirrored in vivo, with pERK detection in the *Pax6^Le^-Cre;Shp2^f/CS^* lens but not in the *Pax6^Le^-Cre;Shp2^f/f^* mutants (*Figure 6I*). Consequently, lens epithelial cells in *Pax6^Le^-Cre;Shp2^f/f^* mutants migrated to the posterior pole with reduced p57 and Jag1 expression, indicating impaired differentiation, but *Pax6^Le^-Cre;Shp2^f/CS^* lens epithelial cells showed proper p57-mediated cell cycle exit at the lens equator and initiated timely expression of Jag1 (*Figure 6J*). However, these lenses showed normal proliferation (Ki67) but increased cell death (TUNEL), resulting in a smaller size (*Figure 6—figure supplement 1*). Moreover, the development of the FGF signaling-sensitive lacrimal gland was blocked in both *Pax6^Le^-Cre;Shp2^f/f^* and *Pax6^Le^-Cre;Shp2^f/CS^* mutants, a more pronounced effect than the modest reduction in lacrimal gland buds observed in *Shp2^YF/YF^* mutants (*Figure 6I*). This suggests that inhibiting Shp2's phosphatase activity more significantly affects FGF signaling compared to obstructing its adaptor function, yet doesn't completely abolish FGF signaling.

## Shc1 cooperates with Frs2 and Shp2 to promote lens development

The mild lens defects observed in *Shp2* mutants lacking either adaptor or phosphatase function led us to investigate alternative mechanisms for Grb2 recruitment to the FGFR complex. Given the distinct lens phenotypes in *Pax6^Le^-Cre;Fgfr1^f/f^;Fgfr2^f/LR^* and *Pax6^Le^-Cre;Fgfr1^f/ΔFrs^;Fgfr2^f/f^* mutants (*Figure 7A*), we hypothesized that FGF might activate unidentified factor(s) in *Fgfr1^f/f^;Fgfr2^f/LR^* MEF cells but not in *Fgfr1^f/ΔFrs^;Fgfr2^f/f^* cells following the excision of flox alleles by Cre-expressing adenovirus, mirroring the observed pattern of pERK activation. Interestingly, FGF stimulation was ineffective in raising pFrs2 and pShp2 levels in both sets of MEF cells and did not alter the phosphorylation states of Crk and Gab1, both recognized adaptors in FGF signaling (*Collins et al., 2018*; *Hadari et al., 2001*; *Li et al., 2014*). However, a key difference emerged – Shc phosphorylation was lost in *Fgfr1^f/ΔFrs^;Fgfr2^f/f^* cells but persisted in *Fgfr1^f/f^;Fgfr2^f/LR^* mutants (*Figure 7A–B*). This observation was further supported by in vivo data, which showed that pShc was detectable in both wild-type and *Pax6^Le^-Cre;Fgfr1^f/f^;Fgfr2^f/LR^* lens vesicles, but not in *Pax6^Le^-Cre;Fgfr1^f/ΔFrs^;Fgfr2^f/f^* mutants (*Figure 7C*, arrowhead). These findings suggest that Shc can be activated by the FGFR independently of its Frs2 binding site, potentially serving as an alternate route for Grb2's engagement.

Among the four *Shc* genes present in the mammalian genome, *Shc1* is the most abundant in the lens (iSyTE lens gene expression database). This led us to ablate *Shc1* in the lens to determine its function in FGF signaling. However, *Pax6^Le^-Cre;Shc1^f/f^* lenses displayed only minor reductions in pERK staining and size, suggesting potential redundancy with other Shc proteins or compensatory mechanisms involving Frs2 and Shp2 (*Figure 7D*). To investigate this further, we created compound mutants involving these genes. We have previously reported that the deletion of *Frs2* or *Shp2* alone led to a modest diminution in lens size, akin to the *Shc1* deletion effect. However, the combined knockout of *Frs2* and *Shp2* (*Pax6^Le^-Cre;Frs2^f/f^;Shp2^f/f^*) resulted in hollow lens vesicles, highlighting a synergistic interaction between these two genes (*Li et al., 2014*). While the *Pax6^Le^-Cre;Frs2^f/f^;Shc1^f/f^*

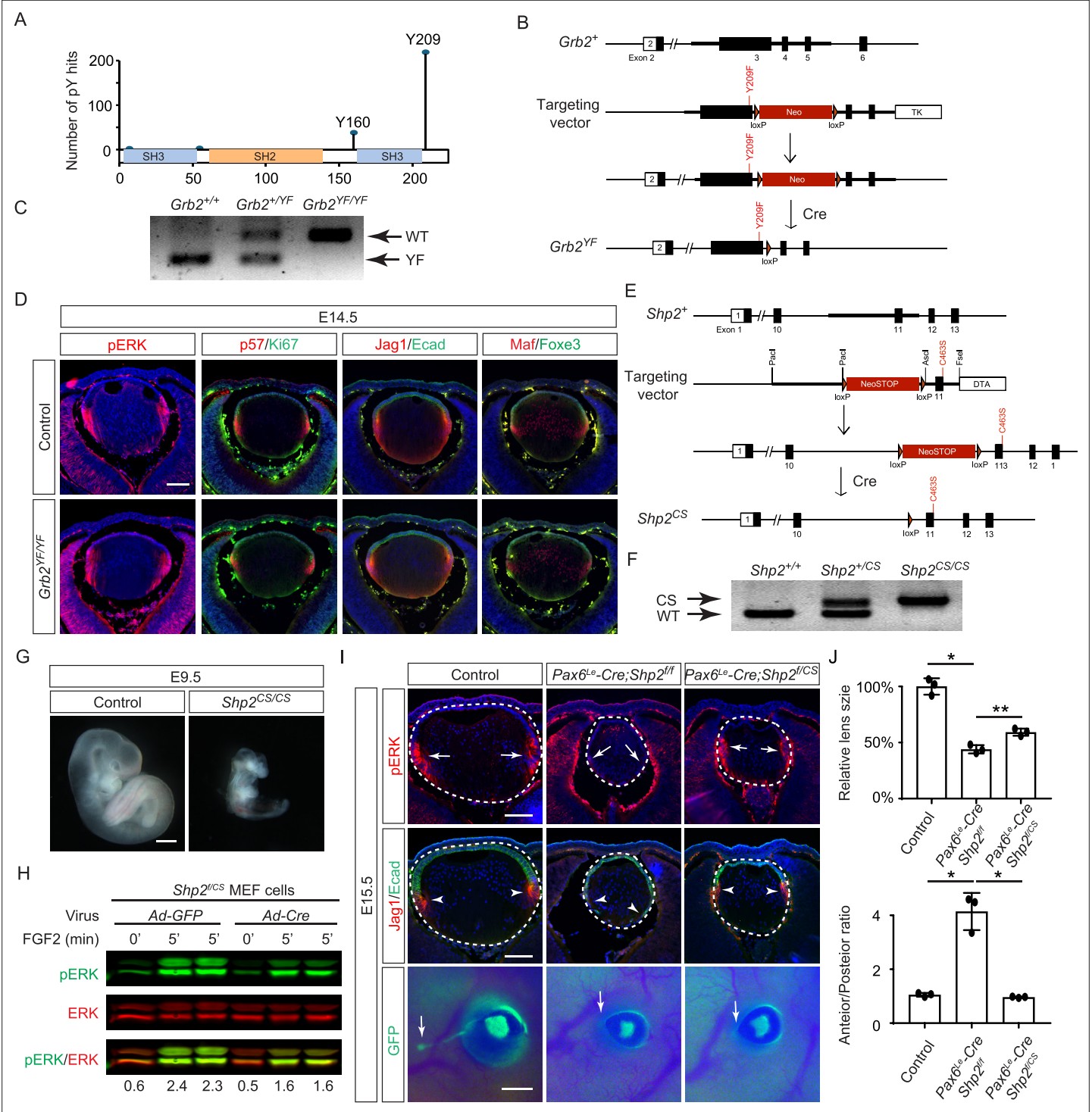

**Figure 6.** Shp2 phosphatase activity is partially required for fibroblast growth factor (FGF) signaling independently of Grb2 dephosphorylation. (**A**) PhositePlus database indicates that Grb2 predominantly undergoes phosphorylation at Y209 and less frequently at Y160. (**B**) The *Grb2^YF* allele was constructed by homologous recombination to integrate a *loxP*-flanked *Neo* cassette and a Y209F point mutation into the *Grb2* locus. (**C**) PCR genotyping confirmed the presence of the *Grb2^YF* allele. (**D**) *Grb2* mutant lens typical expression patterns of pERK, cell cycle markers p57 and Ki67 and differentiation markers Foxe3, Maf, and Jag1. (**E**) The *Shp2^CS* allele was generated by inserting the C459S mutation into the Shp2 locus by homologous recombination, followed by Cre-mediated removal of the *loxP*-flanked *Neo* cassette. (**F**) *Shp2^CS* allele validated by PCR genotyping. (**G**) *Shp2^CS/CS* mutants exhibited growth retardation and died at E9.5. (**H**) *Shp2^f/CS* MEF cells retained a significant capacity to activate pERK upon FGF stimulation after Cre virus infection. (**I**) *Pax6^Le-Cre;Shp2^f/f* mutants exhibited loss of pERK and Jag1 staining at the lens transition zone (arrowheads), which also shifted

*Figure 6 continued on next page*

*Figure 6 continued*

posteriorly. *Pax6^Le^-Cre;Shp2^f/CS* mutant lens, in contrast, maintained staining at the equatorial region. Notably, both mutant types lacked lacrimal gland buds (arrows). (**J**) Quantification of the lens size. One-way ANOVA, n=3, *p<0.0001, **p<0.001. (**K**) Quantification of the lens perimeter spanning the anterior epithelium versus that of the posterior lens fiber. One-way ANOVA, n=3, *p<0.001. Scale bars:50 μm.

The online version of this article includes the following source data and figure supplement(s) for figure 6:

**Source data 1.** Original files for PCR analysis displayed in *Figure 6C*.

**Source data 2.** Original image corresponding to *Figure 6*, panel C, with the relevant lanes are outlined in yellow.

**Source data 3.** Original files for PCR analysis displayed in *Figure 6F*.

**Source data 4.** Original image corresponding to *Figure 6*, panel F, with the relevant lanes are outlined in yellow.

**Source data 5.** Original files for western blot analysis displayed in *Figure 6H*.

**Source data 6.** Original membranes corresponding to *Figure 6*, panel H, with the relevant lanes are outlined in yellow.

**Figure supplement 1.** Cell proliferation and apoptosis in *Shp2^CS^* mutants.

compound mutant did not exhibit as severe abnormalities, introducing the *Shc1* knockout into the *Pax6^Le^-Cre;Frs2^f/f^;Shp2^f/f^* background further diminished the lens size, suggesting that Shc1 contributes an additive role alongside Frs2 and Shp2 in modulating lens development (*Figure 7E–F*). These findings collectively showed that Shc1 functions independently of Frs2 and Shp2 to transmit FGF signaling in lens development.

## Discussion

This study used lens development as a model system to dissect the intricate mechanisms of FGF signaling. By systematically disrupting FGF signaling components, it unveiled a previously unappreciated dependency on precise FGF dosage for each developmental stage. Genetic rescue experiments and targeted manipulation of tyrosine phosphorylation further demonstrated that FGF signaling relies only partially on Frs2 and Shp2 for Grb2 recruitment, ultimately activating Ras and preventing cell death. Contrary to prevailing expectations, while both the adaptor function and the phosphatase activity of Shp2 are vital for embryonic survival, they play a surprisingly modest role in lens development, challenging the current understanding of Shp2 signaling mechanism. Notably, our research suggests that Shc may provide an alternative pathway for Grb2 recruitment and subsequent Ras activation. Although various adaptor proteins like Frs2, Crk, Shb, and Gab have been recognized for their roles in FGF signal transduction, recent findings suggest that mutations in their binding sites on FGF receptors have a significantly lesser impact compared to *Fgfr* null mutations (*Brewer et al., 2015*; *Klint and Claesson-Welsh, 1999*). Our data propose that Shc1 serves as an alternative route for FGF signal transmission, thereby adding a new dimension to our understanding of FGF signaling dynamics.

Lens induction, the pivotal event in eye development, has captivated developmental biologists since Hans Spemann's seminal discovery over a century ago, which identified the optic vesicle's role in triggering the overlying head ectoderm to differentiate into the lens (*Spemann, 1901*). However, the nature of the lens inductive signal has remained elusive (*Makrides et al., 2022*; *Robinson, 2006*). Previous studies suggested that FGF signaling might not be essential, as deleting Fgfr1/2 in the head ectoderm did not affect the expression of the early lens determination gene, Foxe3 (*Garcia et al., 2011*). Contrary to this notion, we have ablated all FGFRs present in the surface ectoderm, which led to a complete loss of pERK, confirming the absence of FGF signaling activity. While the mutant ectoderm still expressed Pax6, this observation may be confounded by the fact that the *Pax6^Le^*-Cre driver used for FGFR deletion depends on Pax6 activity. Importantly, the mutant tissue failed to upregulate Sox2 and Foxe3, the definitive markers of the lens induction. Moreover, the disruption of FGF signaling prevented the apical confinement of F-actin, impeding lens placode invagination driven by apical constriction (*Chauhan et al., 2011*). Considering the concomitant expression of FGF ligands in the optic vesicle, our findings suggest that FGF signaling is an indispensable factor in lens induction.

The role of Shp2 phosphatase in enhancing Receptor Tyrosine Kinase (RTK) signaling pathways is well established, yet its molecular mechanism remains largely unresolved (*Neel et al., 2003*). The prevailing model posits that Shp2 functions by dephosphorylating key tyrosine residues on target proteins, thereby activating Ras signaling. However, it has also been suggested that Shp2 itself can undergo C-terminal phosphorylation, potentially serving as a docking platform for other signaling

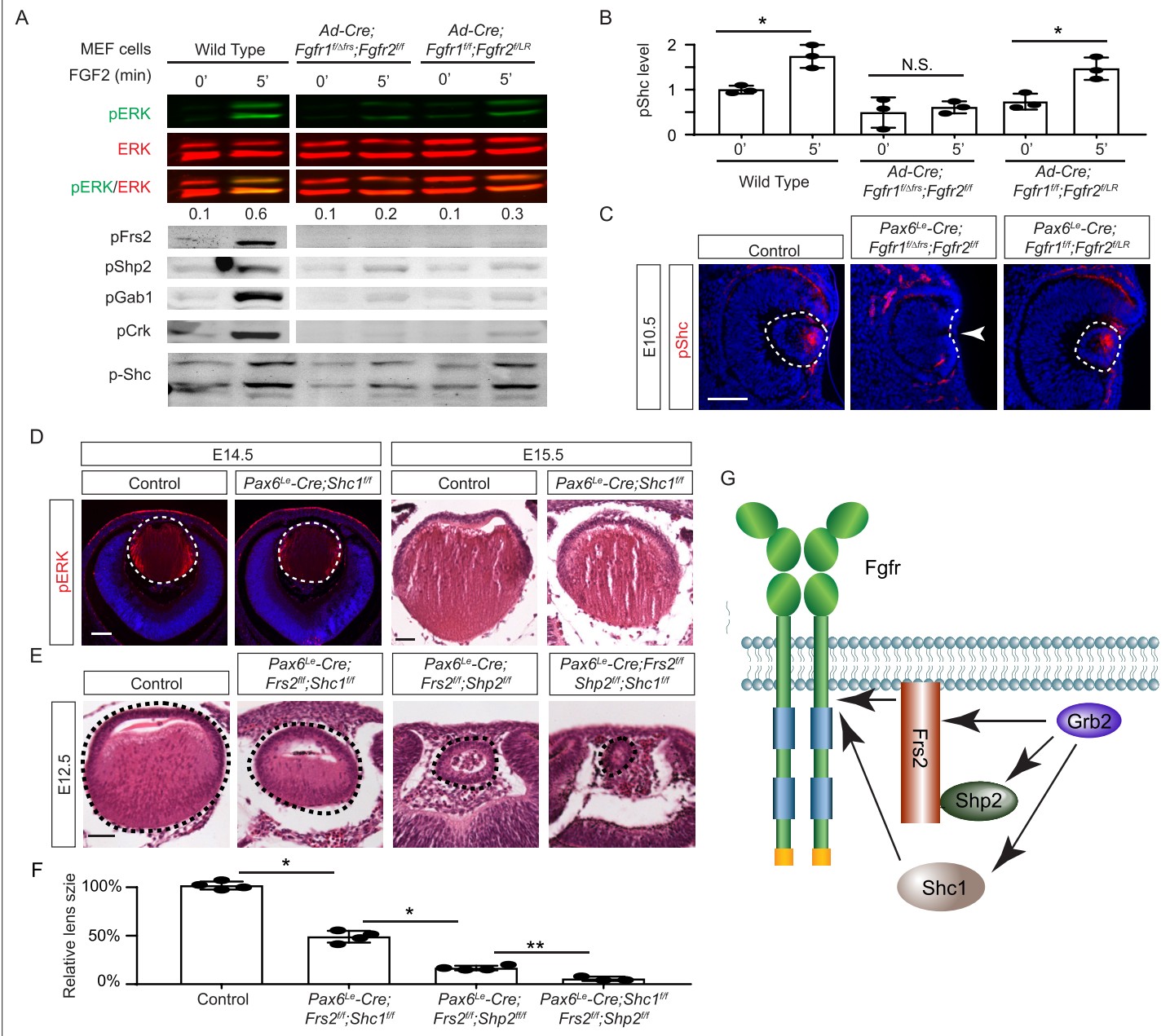

**Figure 7.** Shc1 complements Frs2 and Shp2 in mediating fibroblast growth factor (FGF) signaling in lens development. (**A**) *Fgfr1^{f/f};Fgfr2^{f/LR}* infected with Cre virus showed stronger pERK and pShc activation than *Fgfr1^{f/ΔFrs};Fgfr2^{f/f}* mouse embryonic fibroblast (MEF) cells, despite both losing Frs2, Shp2, Gab1, and Crk phosphorylation. (**B**) Quantification of pShc levels. One-way ANOVA, n=3, *p<0.02. N.S. Not significant. (**C**) pShc staining was lost in *Pax6^{Le}-Cre;Fgfr1^{f/ΔFrs};Fgfr2^{f/f}* mutant lens (arrowhead) but preserved in *Pax6^{Le}-Cre;Fgfr1^{f/f};Fgfr2^{f/LR}* lens. (**D**) *Shc1*-deficient lenses showed a slight decrease in both pERK staining intensity and overall lens size. (**E**) E12.5 *Frs2/Shc1* mutant lenses were smaller than controls, while lenses from *Frs2/Shp2* mutants showed a pronounced hollow vesicle structure, a condition that worsened in *Frs2/Shp2/Shc1* triple mutants. (**F**) Quantification of the lens size. One-way ANOVA, n=4, *p<0.001, **p<0.05. (**G**) Model of FGF signaling network. Frs2 recruits Grb2 directly and indirectly through Shp2, while Shc1 provides an alternate Grb2 recruitment route independent of Frs2. Scale bars:50 μm.

The online version of this article includes the following source data for figure 7:

**Source data 1.** Original files for western blot analysis displayed in *Figure 7A*.

**Source data 2.** Original membranes corresponding to *Figure 7*, panel A, with the relevant lanes are outlined in yellow.

molecules. Through targeted mutagenesis of these putative phosphorylation sites (*Shp2^YF*), we demonstrate the critical role of Shp2 phosphorylation in placental formation and neonatal survival. Biochemical analyses revealed distinct cellular responses to FGF and PDGF stimulation in *Shp2^YF* mutants, suggesting context-dependent functions for these C-terminal modifications, which may explain the narrow phenotypic spectrum associated with the *Shp2^YF* mutation. Intriguingly, the inactivation of Shp2 phosphatase activity via the *Shp2^CS* mutation also resulted in minimal disruption of FGF signaling in lens development, unlike the control *Pax6^Le-Cre;Shp2^f/f* mutant, which showed a significant reduction in pERK levels and subsequent lens differentiation defects. This lack of a robust phenotype cannot be attributed to residual protein activity following Cre-mediated gene deletion due to the systemic nature of the *Shp2^CS* mutation. While Shp2 phosphatase deficiency did cause early embryonic lethality and abrogated the development of the more FGF-sensitive lacrimal gland, the absence of a pronounced lens phenotype calls into question the essentiality of Shp2 phosphatase activity for its overall function. Given the distinct phenotype observed in the *Shp2^YF* mutant, it is plausible that Shp2 fulfills dual roles as both an adaptor and a phosphatase in certain signaling contexts, challenging existing paradigms and inviting further investigation into its multifaceted biological functions.

Previous studies have identified FGFR as a docking station for various signaling adaptor proteins, including Frs2, Plcg, Crk, Grb14, and Shb. In a herculean effort, Soriano and colleagues have eliminated these binding sites, both individually and collectively, but the outcomes were unexpectedly mild compared to the more severe phenotypes observed in the corresponding null mutants (**Brewer et al., 2015**; **Clark and Soriano, 2024**). For instance, whereas the *Fgfr1* null mutant is lethal by E6.5, mutants lacking the ability to bind Frs2, CrkL, and Plcg/Shb/Grb14 survive until E10.5. More strikingly, *Fgfr2* null mutants typically succumb by E10.5 due to widespread organ development failures, yet mutants deficient in these specific binding sites can reach adulthood with minimal apparent defects. The fact that combining *Fgfr1/2* signaling mutations does not mimic the null phenotype further suggests that these mild mutant phenotypes are not simply a result of compensatory actions by other FGF receptors. This discrepancy highlights a crucial gap in our understanding of FGF signaling, implying the existence of unidentified factors that can compensate for the loss of Frs2 and other adaptors. Our study proposes Shc1 as a potential player in this complex signaling web, which is consistent with the observations that Shc1 can be phosphorylated in association with FGF receptors at sites known to facilitate Grb2 binding (**Klint et al., 1995**; **Schüller et al., 2008**), indicating an alternative route for signal propagation. Previous studies have shown that Shc proteins are predominantly expressed in lens epithelial cells and downregulated in lens fiber cells of the chick lens. Despite this downregulation, Shc becomes associated with α6A integrin in the lens fiber zone (**Walker et al., 2002**). However, the functional significance of Shc's abundant expression in the lens epithelial zone remained unclear. In our study, we found that *Shc1* knockout models exhibited relatively mild lens phenotypes at E15.5 during lens differentiation, possibly due to functional redundancy among Shc family proteins. Importantly, we discovered that Shc phosphorylation is dependent on FGF signaling during lens vesicle development. Furthermore, simultaneous deletion of *Shc1*, *Frs2*, and *Shp2* exacerbated lens development defects, resulting in a hollow lens vesicle significantly smaller than that observed in Frs2/Shp2 double mutants. These results demonstrate that Shc participates in FGF signaling in lens epithelial cells during lens vesicle development. These findings point towards a robust and adaptable FGF signaling network, capable of engaging alternative pathways through Frs2, Shc, and other adaptor proteins, thereby maintaining its function despite significant genetic disruptions. These insights underscore the complexity and resilience of cellular signaling networks, understanding which is important for developing strategies to manipulate them in developmental and disease contexts.

## Materials and methods

**Key resources table**

| Reagent type (species) or resource | Designation | Source or reference | Identifiers | Additional information |
|---|---|---|---|---|
| Strain (Adenovirus) | Ade-GFP | Iowa viral vector core | Cat#VVC-U of Iowa-4 | |
| Strain (Adenovirus) | Ade-cre | Iowa viral vector core | Cat#VVC-U of Iowa-1174 | |

*Continued on next page*

*Continued*

| Reagent type (species) or resource | Designation | Source or reference | Identifiers | Additional information |
|---|---|---|---|---|
| Genetic reagent (*M. musculus*) | Fgfr1<sup>DFrs</sup> | *Hoch and Soriano, 2006* | RRID:MGI:3620075 | Dr. Raj Ladher (RIKEN Kobe Institute-Center for Developmental Biology, Kobe, Japan) |
| Genetic reagent (*M. musculus*) | Fgfr2<sup>LR</sup> | *Eswarakumar et al., 2006* | RRID:MGI:3699819 | Dr. Jacob V.P. Eswarakumara (Yale University School of Medicine, New Haven, CT) |
| Genetic reagent (*M. musculus*) | Fgfr3<sup>flox</sup> | *Su et al., 2010* | RRID:MGI:4459831 | Dr. Xin Sun (University of California at San Diego, San Diego, CA) |
| Genetic reagent (*M. musculus*) | Fgfr4<sup>-/-</sup> | *Weinstein et al., 1998* | RRID:MGI:3653043 | Dr. Chu-Xia Deng (National Institute of Health, Bethesda, MD) |
| Genetic reagent (*M. musculus*) | Frs2<sup>flox</sup> | *Lin et al., 2007* | RRID:MGI:3768915 | Fen Wang (Texas A&M, Houston, TX) |
| Genetic reagent (*M. musculus*) | Grb2<sup>flox</sup> | *Ackermann et al., 2011* | RRID:MGI:4949890 | Dr. Lars Nitschke (University of Erlangen-Nürnberg, Erlangen, Germany) |
| Genetic reagent (*M. musculus*) | P6 5.0 lacZ | *Makarenkova et al., 2000* | | Dr. Paul A. Overbeek (Baylor College of Medicine, Houston, TX) |
| Genetic reagent (*M. musculus*) | Shc1<sup>flox</sup> | *Hardy et al., 2007* | RRID:MGI:3716783 | Tony Pawson (University of Toronto, Ontario, Canada) |
| Genetic reagent (*M. musculus*) | Shp2<sup>flox</sup> | *Zhang et al., 2004* | RRID:MGI:3522138 | Gen-sheng Feng (UCSD, Sad Diego, CA) |
| Genetic reagent (*M. musculus*) | Kras<sup>G12D</sup> | *Tuveson et al., 2004* | RRID:MGI:3044567 | Mouse Models of Human Cancers Consortium (MMHCC) Repository at National Cancer Institute |
| Genetic reagent (*M. musculus*) | Pax6<sup>Le</sup>-Cre | *Ashery-Padan et al., 2000* | RRID:MGI:3045795 | Richard Lang (Children's Hospital Research Foundation, Cincinnati, OH) |
| Genetic reagent (*M. musculus*) | Fgfr1<sup>flox</sup> | Jackson Lab | RRID:MGI:3713779 | |
| Genetic reagent (*M. musculus*) | Bax<sup>flox/flox</sup>;Bak<sup>KO/KO</sup> | Jackson Lab | RRID:IMSR_JAX:006329 | |
| Genetic reagent (*M. musculus*) | Sox2Cre | Jackson Lab | Stock #: 008454 RRID:IMSR_JAX:008454 | |
| Genetic reagent (*M. musculus*) | Fgfr2<sup>flox</sup> | *Yu et al., 2003* | RRID:MGI:3044690 | Dr. David Ornitz, Washington University Medical School, St Louis, MO |
| Genetic reagent (*M. musculus*) | Shp2<sup>YF</sup> | This study | | Described in Methods and Materials. Mice are available upon request. |
| Genetic reagent (*M. musculus*) | Shp2<sup>CS</sup> | This study | | Described in Methods and Materials. Mice are available upon request. |
| Genetic reagent (*M. musculus*) | Grb2<sup>YF</sup> | This study | | Described in Methods and Materials. Mice are available upon request. |
| Cell line | Mouse embryonic fibroblast cells | Made from E13.5 embryos | | Primary cells |
| Antibody | Rabbit monoclonal anti-phospho-ERK1/2 | Cell Signaling | Cat#4370 RRID:AB_2315112 | IHC (1:200) |
| Antibody | Rabbit polyclonal anti-phospho-mTOR | Cell Signaling | Cat#2971 RRID:AB_330970 | IHC (1:100) |
| Antibody | Rabbit monoclonal anti-phospho-pS6 | Cell Signalling | Cat#5364 RRID:AB_10694233 | IHC (1:300) |
| Antibody | Rabbit polyclonal anti-phospho-Shc | Cell Signalling | Cat#2434 RRID:AB_10841301 | IHC (1:100) WB (1:500) |
| Antibody | Rabbit monoclonal anti-phospho-MEK1/2 | Cell Signalling | Cat#2338 RRID:AB_490903 | IHC (1:200) |
| Antibody | Rabbit monoclonal anti-phospho-Smad1/5/9 | Cell Signalling | Cat#13820 RRID:AB_2493181 | IHC (1:500) |

*Continued on next page*

*Continued*

| Reagent type (species) or resource | Designation | Source or reference | Identifiers | Additional information |
|---|---|---|---|---|
| Antibody | Rabbit Polyclonal Anti- Phospho-FRS2 (Y436) | Cell Signalling | Cat#3861 RRID:AB_2231950 | IHC (1:500) |
| Antibody | Rabbit polyclonal Anti-phospho-Shp2 (Tyr542) | Cell Signalling | Cat#3751 RRID:AB_330825 | WB (1:1000) |
| Antibody | Mouse monoclonal Anti-pERK(E-4) | Santa Cruz Biotechnology | Cat#7383 RRID:AB_627545 | WB (1:2000) |
| Antibody | Rabbit monoclonalAnti-phospho-Gab1 (Y307) | Cell Signalling | Cat#3233 RRID:AB_2107683 | WB (1:500) |
| Antibody | Rabbit anti-phospho-crk II | Cell Signalling | Cat#3491 RRID:AB_2229920 | WB (1:1000) |
| Antibody | Rabbit monoclonal anti-LEF1 | Cell Signalling | Cat#2230 RRID:AB_823558 | IHC (1:300) |
| Antibody | Rabbit monoclonal anti-Cleaved caspase3 | Cell Signalling | Cat#9664 RRID:AB_2070042 | IHC (1:100) |
| Antibody | Mouse monoclonal anti-cyclin D1 | Cell Signalling | Cat#2926 RRID:AB_2070400 | IHC (1:100) |
| Antibody | Rabbit monoclonal anti-cyclin D1 | Cell Signalling | Cat#2978 RRID:AB_2259616 | IHC (1:100) |
| Antibody | Mouse monoclonal anti-cyclin D3 | Cell Signalling | Cat#2936 RRID:AB_2070801 | IHC (1:100) |
| Antibody | Rabbit monoclonal anti-N-cadherin | Cell Signalling | Cat#13116 RRID:AB_2687616 | IHC (1:300) |
| Antibody | Rabbit polyclonal anti-Fibronectin | Millipore Sigma | Cat#AB2033 | IHC (1:300) |
| Antibody | Mouse monoclonal anti-N-cadherin | BD | Cat#610920 RRID:AB_2077527 | IHC (1:200) |
| Antibody | Rabbit polyclonal anti-Maf | Santa Cruz Biotechnology | Cat#sc-7866 RRID:AB_638562 | IHC (1:200) |
| Antibody | Rabbit Polyclonal anti-Jag1 | Santa Cruz Biotechnology | Cat#sc-6011 RRID:AB_649689 | IHC (1:200) |
| Antibody | Mouse monoclonal anti-Foxe3 | Santa Cruz Biotechnology | Cat#sc-377465 | IHC (1:100) |
| Antibody | Mouse monoclonal anti-Ecadherin | BD | Cat#610181 RRID:AB_397580 | IHC (1:200) |
| Antibody | Mouse monoclonal anti-Ki67 | BD | Cat#550609 RRID:AB_393778 | IHC (1:200) |
| Antibody | Chicken Polyclonal anti-GFP | Aves Labs | Cat#GFP-1010 RRID:AB_2307313 | IHC (1:200) |
| Antibody | Rabbit monoclonal anti-p57 | Abcam | Cat# 75947 | IHC (1:200) |
| Antibody | Rabbit Polyclonal anti-Prox1 | Covance | Cat#PRB-238C RRID:AB_291595 | IHC (1:200) |
| Antibody | Rabbit Polyclonal anti-Pax6 | Covance | Cat#PRB-278P RRID:AB_291612 | IHC (1:300) |
| Antibody | Rat monoclonal anti-Sox2 | Thermo Fisher | Cat#14-9811-82 RRID:AB_11219471 | IHC (1:200) |
| Antibody | Rabbit Polyclonal anti-α-crystallin | Sam Zigler (National Eye Institute) | | IHC (1:5000) |
| Antibody | Rabbit Polyclonal anti-γ-crystallin | Sam Zigler (National Eye Institute) | | IHC (1:5000) |
| Peptide, recombinant protein | Recombinant murine FGF2 | ScienCell | Cat# 124–02 | |
| Peptide, recombinant protein | PDGF-AA | R&D | Cat#221-aa | |
| Chemical compound | Phallodin-488 | Life Technology | A12379 | IHC (1:200) |
| Commercial assay Kit | In situ cell death detection kit | Roche | Cat# 1168479510 | |

## Mice

All procedures related to animal care and experimentation were conducted in adherence to the protocols and guidelines approved by the Institutional Animal Care and Use Committee at Columbia University. We obtained *Fgfr1*$^{\Delta Frs}$ from Dr. Raj Ladher (RIKEN Kobe Institute-Center for Developmental Biology, Kobe, Japan) (**Hoch and Soriano, 2006**), *Fgfr2*$^{LR}$ from Dr. Jacob V.P. Eswarakumara (Yale University School of Medicine, New Haven, CT) (**Eswarakumar et al., 2006**) and *Fgfr2*$^{flox}$ from Dr. David Ornitz (Washington University Medical School, St Louis, MO) (**Yu et al., 2003**), *Fgfr3*$^{flox}$ from Dr. Xin Sun (University of California San Diego, La Jolla, CA) (**Su et al., 2010**), *Fgfr4*$^{-/-}$ from Dr. Chu-Xia

Deng (National Institute of Health, Bethesda, MD) (*Weinstein et al., 1998*), *Frs2^flox* from Fen Wang (Texas A&M, Houston, TX) (*Lin et al., 2007*), *Grb2^flox* from Dr. Lars Nitschke (University of Erlangen-Nürnberg, Erlangen, Germany) (*Ackermann et al., 2011*), *Pax6^Le-Cre* from Richard Lang (Children's Hospital Research Foundation, Cincinnati, OH) (*Ashery-Padan et al., 2000*), *Pax6^5.0 lacZ* (*Pax6-LacZ*) reporter transgenic mice from Dr. Paul A. Overbeek (Baylor College of Medicine, Houston, TX) (*Makarenkova et al., 2000*), *Shc1^flox* from Tony Pawson (University of Toronto, Ontario, Canada) (*Hardy et al., 2007*), *Shp2^flox* from Gen-sheng Feng (UCSD, Sad Diego, CA) (*Zhang et al., 2004*). *LSL-Kras^G12D* mice was obtained from the Mouse Models of Human Cancers Consortium (MMHCC) Repository at National Cancer Institute (*Tuveson et al., 2004*). *Bax^flox/flox;Bak^KO/KO* (Stock No: 006329), *Fgfr1^flox* (Stock No: 007671), *Sox2Cre* (Stock No: 008454) mice were obtained from Jackson Laboratory. Animals were maintained on mixed genetic backgrounds. In all conditional knockout experiments, mice were maintained on a mixed genetic background and *Pax6^Le-Cre* only or *Pax6^Le-Cre* and heterozygous flox mice were used as controls. Mouse maintenance and experimentation were performed according to protocols approved by the Columbia University Institutional Animal Care and Use Committee.

*Shp2^YF*, *Shp2^CS* and *Grb2^YF* targeting vectors were constructed using the recombineering method from C57BL/6 Bac clones (RP23-257E17 for *Shp2*, P23-2814 for *Grb2*, BACPAC Resources Center at Children's Hospital Oakland Research Institute) (*Carbe et al., 2012*). The *Shp2^YF* vector includes a neomycin resistance (*Neo*) cassette bordered by *loxP* sites, along with exon 14 of the *Shp2* gene harboring Y542F mutations and exon 15 with Y580F mutations. The *Shp2^CS* vector comprises a *NeoSTOP* cassette encased by *loxP* sites and exon 11 of the *Shp2* gene with the C463S mutation. Similarly, the *Grb2^YF* vector contains a *NeoSTOP* cassette flanked by *loxP* sites and exon 3 of the *Grb2* gene with the Y209F mutation. These targeting constructs, once linearized, were introduced into C57BL/6 and 129 hybrid ES cells via electroporation. *Shp2^YF* recombinant clones were screened by Southern blot analysis with 5' and 3' external probes after restriction digestion with *EvoR V*, while *Grb2^YF* and *Shp2^CS* clones were identified by long-range PCR before being injected into C57BL/6 blastocysts. Chimeras were further bred with C57BL/6 mice for germline transmission, verified through PCR genotyping with specific primers for each mutation: *Grb2^YF* F: 5'- TGGGGGTCAAAGTCAAAGAG –3'; *R*: 5'- CGGAGGGAGTGAGGTATGAG –3' (wild type: 179 bp, mutant: 270 bp), *Shp2^YF* F: 5'- AAAA AGAGGCTGCTCTGCAC –3'; *R*: 5'- TCTGCAGAATGAGGGAGGAC –3' (wild type: 195 bp, mutant: 250 bp) and *Shp2^YF* F: 5'- TGGGAAGACAGACTGCAGTC-3'; *R*: 5'- GAAGGAGCACCTGCCTGTTA-3' (wild type: 180 bp, mutant: 210 bp). The Neo cassette was subsequently excised by breeding with an *EIIa-cre* transgenic line (stock number 003724, Jackson Laboratory, Bar Harbor, ME).

## Database analysis

The cumulative references for each phosphorylation site, derived from both low-throughput (LTP) and high-throughput (HTP) experiments, were sourced from the PhosphositePlus database (phosphosite.org) and graphically represented along the amino acid sequence. The expression data for lens genes from embryonic days 10.5–12.5 was extracted from the iSyTE database (https://research.bioinformatics.udel.edu/iSyTE/ppi/expression.php) and visualized as heatmaps to illustrate the variations in expression levels over time.

## Histology and immunohistochemistry

Histology and immunohistochemistry were performed on the paraffin and cryosections as previously described (*Carbe et al., 2012*; *Carbe and Zhang, 2011*). For hematoxylin and eosin (H&E) staining, 10 μM paraffin sections underwent deparaffinization with histosol wash, rehydration through decreasing concentrations of ethanol solutions, and final washing in water. The slides were immersed in hematoxylin for 3 min, followed by a 10–15 min wash with tap water. Subsequently, they were decolorized with 1% acid alcohol for 30 s before treatment with eosin for 1 min. Samples were dehydrated through increasing ethanol concentrations, transferred to histosol, and mounted using a Permount mounting medium. For X-gal staining, the mouse lacrimal gland was exposed by dissecting away the periocular skin and fixed in 4% paraformaldehyde at 4 °C overnight.

The antibodies used are phospho-ERK1/2 (#4370), phospho-mTOR (#2971), phospho-S6 (#5364), phospho-Shc (#2434), phospho-MEK1/2 (#2338), phospho-Smad1/5/9 (#13820), LEF1 (#2230), Cleaved caspase3 (#9662), cyclin D1 (#2926, discontinued), cyclin-D1 (#2978), cyclin-D3(#2936), N-cadherin (#13116) (all from Cell Signaling Technology), Fibronectin (AB2033) (from Millipore), Foxe3

(#377465), Jag1(#6011), Maf (#7866) (all from Santa Cruz), E-cadherin (#610181) and Ki67 (#550609) (both from BD Pharmingen), GFP (GFP-1010) (from Aves Labs), p57 (#75947) (from Abcam), Prox1 (PRB-238C) and Pax6 (PRB-278P) (both from Covance), Sox2 (14-9811-82) (from Thermo Fisher). Antibodies against α- and γ-crystallins were kindly provided by Sam Zigler (National Eye Institute, Bethesda, MD).

Phospho-ERK, phopho-MEK, phospho-mTOR, phosphor-Shc and phospho-Akt staining was amplified using a Tyramide Signal Amplification kit (TSA Plus System, PerkinElmer Life Sciences, Waltham, MA). Alexa Fluor secondary antibodies and Alexa Fluor 488 phalloidin (A-12379) were ordered from Invitrogen. TUNEL staining was performed following the in situ cell death detection kit (Roche Applied Science, Indianapolis, IN). All commercial antibodies were validated by vendors. At least three embryos of each genotype were stained for each marker.

## Cell culture and western blot

Primary mouse embryonic fibroblast (MEF) cells were generated according to a previously published protocol and cultured in Dulbecco's modified Eagle's medium (DMEM) supplemented with 10% FBS and 1% penicillin-streptomycin. Adenovirus expressing Cre recombinase (Ad-cre) (Gene Transfer Vector Core, University of Iowa, IA.) were applied to MEF cells carrying flox alleles for 5 d to achieve gene deletion, while adenovirus expressing GFP (Ad-GFP) served as a control treatment. To assess growth factor responses, MEF cells were starved overnight and then treated with either FGF2 (50 ng/µl) or PDGFA (20 ng/µl) (from R&D systems) for 5 min. Cells were immediately washed with cold PBS and harvested in ice-cold CelLytic buffer (C2978, Sigma-Aldrich, St.Louis, MO) supplemented with protease and phosphatase inhibitor cocktails (Pierce, Rockford, IL). Extracted proteins were subjected to standard western blot analysis. The antibodies used include ERK1/2 (#4695), phospho-Shp2 (#15543), phospho-Crk(#3491), phospho-Gab1(#3234), phopho-Frs2 (#3861), phospho-Shc (#2434) (all from Cell Signaling Technology), along with phospho-ERK1/2 (#7383) from Santa Cruz Biotechnology.

## Quantification and statistical analysis

The relative lens sizes were measured using Image J and normalized against the control. The anterior/posterior lens ratio was determined by comparing the length of the anterior epithelium, as measured in Image J, to the length of the posterior lens boundary. The percentage of TUNEL and cleaved-caspase3 positive cells were normalized against the total number of DAPI-positive cells. Statistical analysis was performed using GraphPad Prism 7. Sample sizes were not predetermined. Data represent mean ± s.d. Statistical differences between two groups were assessed using an unpaired, two-tailed t-test, while comparisons among three or more groups employed one-way ANOVA followed by Tukey's multiple comparison test.

## Acknowledgements

The authors thank Drs. Ruth Ashery-Padan, Chu-Xia Deng, Jacob V.P. Eswarakumara, Gen-sheng Feng, Raj Ladher, Richard Lang, Lars Nitschke, David Ornitz, Paul A Overbeek, Tony Pawson, Philippe Soriano, Xin Sun, Fen Wang for mice and Lin Gan for generating the Shp2CS strain. The work was supported by grants from NIH (R01EY017061, R01EY018868, and R01EY025933 to X Z). Q W is supported by a Pathway to Independence Award (K99EY032171). The Columbia Ophthalmology Core Facility is supported by NIH Core grant 5P30EY019007 and unrestricted funds from Research to Prevent Blindness (RPB).

## Additional information

### Funding

| Funder | Grant reference number | Author |
| --- | --- | --- |
| National Eye Institute | R01EY017061 | Xin Zhang |
| National Eye Institute | EY018868 | Xin Zhang |

| Funder | Grant reference number | Author |
| --- | --- | --- |
| National Eye Institute | R01EY025933 | Xin Zhang |
| National Eye Institute | K99EY032171 | Qian Wang |

The funders had no role in study design, data collection and interpretation, or the decision to submit the work for publication.

## Author contributions
Qian Wang, Conceptualization, Data curation, Formal analysis, Funding acquisition, Investigation, Methodology, Writing – original draft, Writing – review and editing; Hongge Li, Yingyu Mao, Conceptualization, Data curation, Formal analysis, Investigation, Methodology, Writing – review and editing; Ankur Garg, Eun Sil Park, Yihua Wu, Alyssa Chow, John Peregrin, Investigation, Writing – review and editing; Xin Zhang, Conceptualization, Resources, Data curation, Formal analysis, Supervision, Funding acquisition, Investigation, Methodology, Writing – original draft, Project administration, Writing – review and editing

## Author ORCIDs
Qian Wang ⓘ http://orcid.org/0000-0002-6166-3081
Hongge Li ⓘ https://orcid.org/0000-0003-0237-3225
Xin Zhang ⓘ https://orcid.org/0000-0001-5555-0825

## Ethics
Mouse maintenance and experimentation were performed according to protocols (AABT3653) approved by Columbia University Institutional Animal Care and Use Committee.

Reviewer #2 (Public review): https://doi.org/10.7554/eLife.103615.4.sa1
Author response https://doi.org/10.7554/eLife.103615.4.sa2

---

# Additional files

## Supplementary files
MDAR checklist

## Data availability
All data generated or analysed during this study are included in the manuscript and supporting files.

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
