## [Editor Report · eLife Assessment]

This **fundamental** article significantly advances our understanding of FGF signalling, and in particular highlights the complex modifications affecting this pathway. The evidence for the authors' claims is **convincing**, combining state of the art conditional gene deletion in the mouse lens with histological and molecular approaches. This work should be of great interest to molecular and developmental biologists beyond the lens community.

---

## [Referee Report · Reviewer #2 (Public review)]

Summary:

In this paper, the authors first examined lens phenotypes in mice with Le-Cre-mediated knockdown (KD) of all the four FGFR (FGFR1-4), and found that pERK signals, Jag1 and foxe3 expression are absent or drastically reduced, indicating that FGF signaling is essential for lens induction. Next, the authors examined lens phenotypes of FGFR1/2-KD mice and found that lens fiber differentiation is compromised, and that proliferative activity and cell survival are also compromised in lens epithelium. Interestingly, Kras activation rescues defects in lens growth and lens fiber differentiation in FGFR1/2-KD mice, indicating that Ras activation is a key step for lens development, downstream of FGF signaling. Next, the authors examined the role of Frs2, Shp2 and Grb2 in FGF signaling for lens development. They confirmed that lens fiber differentiation is compromised in FGFR1/3-KD mice combined with Frs2-dysfunctional FGFR2 mutants, which is similar to lens phenotypes of Grb2-KD mice. However, lens defects are milder in mice with Shp2YF/YF and Shp2CS mutant alleles, indicating that involvement of Shp2 is limited for the Grb2 recruitment for lens fiber differentiation. Lastly, the authors showed new evidence on the possibility that another adapter protein, Shc1, promotes Grb2 recruitment independent of Frs2/Shp2-mediated Grb2 recruitment.

Strengths:

Overall, the manuscript provides valuable data on how FGFR activation leads to Ras activation through the adapter platform of Frs2/Shp2/Grb2, which advances our understanding on complex modification of FGF signaling pathway. The authors applied a genetic approach using mice, whose methods and results are valid to support the conclusion. The discussion also well summarizes the significance of their findings.

Weaknesses:

The authors found that the new adaptor protein Shc1 is involved in Grb2 recruitment in response to FGF receptor activation. However, the main data on Shc1 are only histological sections and statistical evaluation of lens size. Cellular-level evidence on Shc1 makes the authors' conclusion more convincing.

Comments on latest version:

In the 2nd revised version of the manuscript, the authors responded to my recommendation to show the number of biological replicates for Prox1 and αA-crystallin (Fig. 1F) and conductedstatistical analysis for pmTOR, and pS6 (Supplementary figure 1B).

The authors also explained why the animals are no longer available for the additional experiments that I requested. I may understand the situation, but hope that the authors will investigate the cellular-level evidence on Shc1 in more detail and report it maybe as another paper in future.

---

## [Author Response]

The following is the authors’ response to the previous reviews

**Public Reviews:**

**Reviewer #1 (Public review):**
Summary:This manuscript uses the eye lens as a model to investigate basic mechanisms in the Fgf signaling pathway. Understanding Fgf signaling is of broad importance to biologists as it is involved in the regulation of various developmental processes in different tissues/organs and is often misregulated in disease states. The Fgf pathway has been studied in embryonic lens development, namely with regards to its involvement in controlling events such as tissue invagination, vesicle formation, epithelium proliferation and cellular differentiation, thus making the lens a good system to uncover the mechanistic basis of how the modulation of this pathway drives specific outcomes. Previous work has suggested that proteins, other than the ones currently known (e.g., the adaptor protein Frs2), are likely involved in Fgfr signaling. The present study focuses on the role of Shp2 and Shc1 proteins in the recruitment of Grb2 in the events downstream of Fgfr activation.Strengths:The findings reveal that the juxtamembrane region of the Fgf receptor is necessary for proper control of downstream events such as facilitating key changes in transcription and cytoskeleton during tissue morphogenesis. The authors conditionally deleted all four Fgfrs in the mouse lens that resulted in molecular and morphological lens defects, most importantly, preventing the upregulation of the lens induction markers Sox2 and Foxe3 and the apical localization of F-actin, thus demonstrating the importance of Fgfrs in early lens development, i.e. during lens induction. They also examined the impact of deleting Fgfr1 and 2, on the following stage, i.e. lens vesicle development, which could be rescued by expressing constitutively active KrasG12D. By using specific mutations (e.g. Fgfr1ΔFrs lacking the Frs2 binding domain and Fgfr2LR harboring mutations that prevent binding of Frs2), it is demonstrated that the Frs2 binding site on Fgfr is necessary for specific events such as morphogenesis of lens vesicle. Further, by studying Shp2 mutations and deletions, the authors present a case for Shp2 protein to function in a context-specific manner in the role of an adaptor protein and a phosphatase enzyme. Finally, the key surprising finding from this study is that downstream of Fgfr signaling, Shc1 is an important alternative pathway - in addition to Shp2 - involved in the recruitment of Grb2 and in the subsequent activation of Ras. The methodologies, namely, mouse genetics and state-of-the-art cell/molecular/biochemical assays are appropriately used to collect the data, which are soundly interpreted to reach these important conclusions. Overall, these findings reveal the flexibility of the Fgf signaling pathway and it downstream mediators in regulating cellular events. This work is expected to be of broad interest to molecular and developmental biologists.Weaknesses:A weakness that needs to be discussed is that Le-Cre depends on Pax6 activation, and hence its use in specific gene deletion will not allow evaluation of the requirement of Fgfrs in the expression of Pax6 itself. But since this is the earliest Cre available for deletion in the lens, mentioning this in the discussion would make the readers aware of this issue.
**Reviewer #2 (Public review):**
SummaryI have reviewed the revised manuscript submitted by Wang et al., which is entitled "Shc1 cooperates with Frs2 and Shp2 to recruit Grb2 in FGF-induced lens development". In this paper, the authors first examined lens phenotypes in mice with Le-Cre-mediated knockdown (KD) of all four FGFR (FGFR1-4), and found that pERK signals, Jag1 and foxe3 expression are absent or drastically reduced, indicating that FGF signaling is essential for lens induction. Next, the authors examined lens phenotypes of FGFR1/2-KD mice and found that lens fiber differentiation is compromised and that proliferative activity and cell survival are also compromised in lens epithelium. Interestingly, Kras activation rescues defects in lens growth and lens fiber differentiation in FGFR1/2-KD mice, indicating that Ras activation is a key step for lens development, downstream of FGF signaling. Next, the authors examined the role of Frs2, Shp2 and Grb2 in FGF signaling for lens development. They confirmed that lens fiber differentiation is compromised in FGFR1/3-KD mice combined with Frs2-dysfunctional FGFR2 mutants, which is similar to lens phenotypes of Grb2-KD mice. However, lens defects are milder in mice with Shp2YF/YF and Shp2CS mutant alleles, indicating that involvement of Shp2 is limited for the Grb2 recruitment for lens fiber differentiation. Lastly, the authors showed new evidence on the possibility that another adapter protein, Shc1, promotes Grb2 recruitment independent of Frs2/Shp2-mediated Grb2 recruitment.StrengthOverall, the manuscript provides valuable data on how FGFR activation leads to Ras activation through the adapter platform of Frs2/Shp2/Grb2, which advances our understanding on complex modification of FGF signaling pathway. The authors applied a genetic approach using mice, whose methods and results are valid to support the conclusion. The discussion also well summarizes the significance of their findings.WeaknessThe authors found that the new adaptor protein Shc1 is involved in Grb2 recruitments in response to FGF receptor activation. However, the main data on Shc1 are only histological sections and statistical evaluation of lens size. In the revised manuscript, the authors did not answer my major concern that cellular-level data are missing, which is not fully enough to support their main conclusion on the involvement of Shc1 in Grb2 recruitment of FGF signaling for lens development. Since the title of this manuscript is that Shc1 cooperates with Frs2 and Shp2 to recruit Grb2 in FGF-induced lens development, it is important to provide the cellular-level evidence on Shc1.
**Reviewer #3 (Public review):**
Summary:The manuscript entitled "Shc1 cooperates with Frs2 and Shp2 to recruit Grb2 in FGF-induced lens development" by Wang et al., investigates the molecular mechanism used by FGFR signaling to support lens development. The lens has long been known to depend on FGFR-signaling for proper development. Previous investigations have demonstrated the FGFR signaling is required for embryonic lens cell survival and for lens fiber cell differentiation. The requirement of FGFR signaling for lens induction has remained more controversial as deletion of both Fgfr1 and Fgfr2 during lens placode formation does not prevent the induction of definitive lens markers such as FOXE3 or αA-crystallin. Here the authors have used the Le-Cre driver to delete all four FGFR genes from the developing lens placode demonstrating a definitive failure of lens induction in the absence of FGFR-signaling. The authors focused on FGFR1 and FGFR2, the two primary FGFRs present during early lens development and demonstrated that lens development could be significantly rescued in lenses lacking both FGFR1 and FGFR2 by expressing a constitutively active allele of KRAS. They also showed that the removal of pro-apoptotic genes Bax and Bak could also lead to a substantial rescue of lens development in lenses lacking both FGFR1 and FGFR2. In both cases, the lens rescue included both increased lens size and the expression of genes characteristic of lens cells.Significantly the authors concentrated on the juxtamembrane domain, a portion of the FGFRs associated with FRS2. Previous investigations have demonstrated the importance of FRS2 activation for mediating a sustained level of ERK activation. FRS2 is known to associate both with GRB2 and SHP2 to activate RAS. The authors utilized a mutant allele of Fgfr1, lacking the entire juxtamembrane domain (Fgfr1ΔFrs) and an allele of Fgfr2 containing two-point mutations essential for Frs2 binding (Fgfr2LR). When combining three floxed alleles and leaving only one functional allele (Fgfr1ΔFrs or Fgfr2LR) the authors got strikingly different phenotypes. When only the Fgfr1ΔFrs allele was retained, the lens phenotype matched that of deleting both Fgfr1 and Fgfr2. However, when only the Fgfr2LR allele was retained the phenotype was significantly milder, primarily affecting lens fiber cell differentiation, suggesting that something other than FRS2 might be interacting with the juxtamembrane domain to support FGFR signaling in the lens. The authors also deleted Grb2 in the lens and showed that the phenotype was similar to that of the lenses only retaining the Fgfr2LR allele, resulting a failure of lens fiber cell differentiation and decreased lens cell survival. However, mutating the major tyrosine phosphorylation site of GRB2 did not affect lens development. The authors additionally investigated the role of SHP2 in lens development by either deleting SHP2 or by making mutations in the SHP2 catalytic domain. The deletion of the SHP2 phosphatase activity did not affect lens development as severely as total loss of SHP2 protein, suggesting a function for SHP2 outside of its catalytic activity. Although the loss of Shc1 alone has only a slight effect on lens size and pERK activation in the lens, the authors showed that the loss of Shc1 exacerbated the lens phenotype in lenses lacking both Frs2 and Shp2. The authors suggest that SHC1 binds to the FGFR juxtamembrane domain allowing for the recruitment of GRB2 in independently of FRS2.Strengths:(1) The authors used a variety of genetic tools to carefully dissect the essential signals downstream of FGFR signaling during lens development.(2) The authors made a convincing case that something other than FRS2 binding mediates FGFR signaling in the juxtamembrane domain.(3) The authors demonstrated that despite the requirement of both the adaptor function and phosphatase activity of SHP2 are required for embryonic survival, neither of these activities is absolutely required for lens development.(4) The authors provide more information as to why FGFR loss has a phenotype much more severe than the loss of FRS2 alone during lens development.(5) The authors followed up their work analyzing various signaling molecules in the context of lens development with biochemical analyses of FGF-induced phosphorylation in murine embryonic fibroblasts (MEFs).(6) In general, this manuscript represents a Herculean effort to dissect FGFR signaling in vivo with biochemical backing with cell culture experiments in vitro.Weaknesses:(1) The authors demonstrate that the loss of FGFR1 and FGFR2 can be compensated by a constitutive active KRAS allele in the lens and suggest that FGFRs largely support lens development only by driving ERK activation. However, the authors also saw that lens development was substantially rescued by preventing apoptosis through the deletion of BAK and BAX. To my knowledge, the deletion of BAK and BAX should not independently activate ERK. The authors do not show whether ERK activation is restored in the BAK/BAX deficient lenses. Do the authors suggest the FGFR3 and/or FGFR4 provide sufficient RAS and ERK activation for lens development when apoptosis is suppressed? Alternatively, is it the survival function of FGFR-signaling as much as a direct effect on lens differentiation?(2) Do the authors suggest that GRB2 is required for RAS activation and ultimately ERK activation? If so, do the authors suggest that ERK activation is not required for FGFR-signaling to mediate lens induction? This would follow considering that the GRB2 deficient lenses lack a problem with lens induction.(3) The increase in p-Shc is only slightly higher in the Cre FGFR1f/f FGFR2r/LR than in the FGFR1f/Δfrs FGFR2f/f. Can the authors provide quantification?(4) The authors have not shown directly that Shc1 binds to the juxtamembrane region of either Fgfr1 or Fgfr2.
**Recommendations for the authors:**

**Reviewer #2 (Recommendations for the authors):**
In the revised manuscript, the authors have responded to my recommendations to revise the original manuscript, except for three suggestions below.(1) The original recommendation: Results (page 6, line 8): The authors mentioned "we observed .... expression of Foxe3 in ...mutant lens cells (Figure 1E, arrows). However, Foxe3-expressing lens cells are a very small population in Figure 1E. It is important to state the decreased number of Foxe3-expressing lens cells in FGFR1/2 mutants. In addition, I would like to request the authors to show histograms indicating sample size and statistical analysis for marker expression: Foxe3 (Figure 1E), Prox1 and aA-crystallin (Fig. 1F), cyclin D1 and TUNEL (Fig. 1G) and pmTOR and pS6 (Supplementary figure 1B).Author's response: We added a statement indicating that the number of Foxe3-expressing cells is reduced in FGFR1/2 mutants, which is now quantified in Fig. 1H. Quantifications for Cyclin D1 and TUNEL are now shown in Fig. 1I and J, respectively. However, we chose not to quantify Prox1, αA-crystallin, pmTOR, and pS6, as the FGFR1/2 mutants showed no staining for these markers.My recommendation: Although the authors have responded to revise the quantification of Foxe3-expressing cells, Cyclin D1 and TUNEL, they did not conduct statistical analysis of Prox1, αA-crystallin, pmTOR, and pS6, because of absence of these marker signals. I understand that no signal makes statistical analysis no meaningful. However, it is still important to indicate how many the authors repeated experiments to confirm the same result. Please indicate the number of biological replicates or independent experiments in the figure legends, for example "Biological replicates, n=3" or "Three independent experiments show similar results". As for pS6 labeling, there seems to be a weak signal in Supplementary Figure 1B, so please show statistical analysis to indicate its histogram.

We have added the number of biological replicates for Prox1 and αA staining in the legend of Fig.1. The review is correct that there is weak staining of pS6, and also pmTOR. The quantification of pS6 and pmTOR staining are now shown in Supplementary Fig. 1C and D.

(2) The original recommendation: Results (page 6, line 19- page 7, line 6): The authors showed that inducible expression of constitutive active Kras, KrasG12D, using Le-Cre, recovered lens size to the half level of wild-type control. However, in the lens of mice with Le-Cre; FGFR1/2f/f; LSL-KrasG12D, pERK was detected in the most posterior edge of the lens fiber core, whereas pERK was detected in the broader area of the lens in control. Furthermore, pMEK was detected in the whole lens of mice with Le-Cre; FGFR1/2f/f; and LSL-KrasG12D, whereas pMEK was detected only in the lens epithelial cells at the equator. So, the spatial profile of pERK and pMEK expression was different from those of wild-type, although the authors observed that Prox1 and Crystallin expression are normally induced in the lens of mice with Le-Cre; FGFR1/2f/f; LSL-KrasG12D. I wonder whether the lens normally develops in mice with Le-Cre; LSL-KrasG12D? Is the lens growth enhanced in mice with Le-Cre; LSL-KrasG12D? Please add the panels of mice with Le-Cre; LSL-KrasG12D in Figure 2B and 2C. In addition, I wonder whether apoptosis is suppressed in the lens of mice with Le-Cre; FGFR1/2f/f; LSL-KrasG12D?Authors' response: Response: As we previously reported (Developmental Biology 355, 2011, 12-20), Le-Cre; LSL-KrasG12D did not lead to enhanced lens growth. While we agree that including images of Le-Cre; LSL-KrasG12D as controls in Fig. 2B and C and evaluating apoptosis in Le-Cre; FGFR1/2f/f; LSL-KrasG12D mutants would be appropriate, we regretfully no longer have these animals available to conduct these experiments.My recommendation: I would like to suggest the authors conduct these experiments again, because the recovery of lens formation by Bax/Bak KD in Fgfr1/2 KD mice (Fig. 2F) suggests that KrasG12D activates the AKT-mediated cell survival pathway as well as that MEK/MAPK pathway downstream of FGF signaling pathway. Regarding the availability of mouse strains, in general, it is necessary to keep animal strains available for sincere response to reviewers' suggestions. Please clarify why these strains are now not available and justify the reason in the response to reviewers' recommendations.

We acknowledge the reviewer's suggested experiments. However, our research utilized multiple mouse strains that are costly to maintain, a challenge that was exacerbated during and after the COVID-19 pandemic. Unfortunately, we no longer have access to the specific mouse strains required to conduct these additional studies.

(3) The original recommendation: Figures 7E, and 7F: The authors showed that lens morphology and lens size evaluation in genetic combinations: control, Frs2/Shc1 KD, Frs2/Shp2 KD, and Frs2/Shp2/Shc1 KD. However, I would like to request the authors to show more detailed data in these genetic combinations, for example, pERK, foxe3, Maf, Prox1, Jag1, p57, cyclin D3, g-crystallin, and TUNEL.Authors' response: Unfortunately, we no longer have these mutant mice to perform these detailed staining.My recommendation: As I mentioned in the statement on weakness above, it is important to provide the cellular-level evidence to support the main conclusion on the involvement of Shc1 in Grb2 recruitment of FGF signaling for lens development, because this is the main novel finding in this manuscript. Regarding the availability of mouse strains, it is generally necessary to keep animal strains available for sincere response to reviewers' suggestions. Please clarify why these strains are now not available and justify the reason in the response to the reviewers' suggestions.

We regret that we did not anticipate these experiments suggested by the reviewer. Unfortunately, we are unable to perform these studies as we no longer maintain the required mouse strains in our colony.

**Reviewer #3 (Recommendations for the authors):**
The changes made by the authors improved the manuscript. I have no further suggestions.